# Cancer cell survival depends on collagen uptake into tumor-associated stroma

Kuo-Sheng Hsu[1], James M. Dunleavey[1], Christopher Szot[1], Liping Yang[1], Mary Beth Hilton[1,2], Karen Morris[1,2], Steven Seaman[1], Yang Feng[1], Emily M. Lutz[1], Robert Koogle[3], Francesco Tomassoni-Ardori[3], Saurabh Saha[4,9], Xiaoyan M. Zhang[4,10], Enrique Zudaire[1,11], Pradip Bajgain[1], Joshua Rose[5], Zhongyu Zhu[6,12], Dimiter S. Dimitrov[6,13], Frank Cuttitta[1], Nancy J. Emenaker[7], Lino Tessarollo[8] & Brad St. Croix[1] ✉

Collagen I, the most abundant protein in humans, is ubiquitous in solid tumors where it provides a rich source of exploitable metabolic fuel for cancer cells. While tumor cells were unable to exploit collagen directly, here we show they can usurp metabolic byproducts of collagen-consuming tumor-associated stroma. Using genetically engineered mouse models, we discovered that solid tumor growth depends upon collagen binding and uptake mediated by the TEM8/ANTXR1 cell surface protein in tumor-associated stroma. Tumor-associated stromal cells processed collagen into glutamine, which was then released and internalized by cancer cells. Under chronic nutrient starvation, a condition driven by the high metabolic demand of tumors, cancer cells exploited glutamine to survive, an effect that could be reversed by blocking collagen uptake with TEM8 neutralizing antibodies. These studies reveal that cancer cells exploit collagen-consuming stromal cells for survival, exposing an important vulnerability across solid tumors with implications for developing improved anticancer therapy.

The tumor microenvironment (TME), which has stimulated an immense amount of research due to its profound impact on tumor growth, is a complex mixture of stromal cells, including cancer-associated fibroblasts, endothelial cells, pericytes and inflammatory cells, as well as extracellular matrix (ECM) proteins[1]. While new endothelial-lined blood vessels foster tumor expansion by providing cancer cells with critical oxygen and nutrients, several basic questions regarding the role of ECM and other stromal cells in the TME remain.

For example, can a subset of cancer-associated fibroblasts (CAFs) with broad tumor-promoting activity be identified? Can collagen, which is ubiquitous in tumors, be used as an alternative fuel source when nutrients are limited? Is it possible to block collagen stimulatory signals while simultaneously preserving its inhibitory signals? The answers to these questions critically impact the potential for new therapeutic approaches to exploit the TME in a cancer-specific manner.

[1]Tumor Angiogenesis Unit, Mouse Cancer Genetics Program (MCGP), National Cancer Institute (NCI), NIH, Frederick, MD 21702, USA. [2]Basic Research Program, Leidos Biomedical Research Inc., Frederick National Laboratory for Cancer Research (FNLCR), Frederick, MD 21702, USA. [3]MCGP, NCI, Frederick, MD 21702, USA. [4]BioMed Valley Discoveries, Inc, Kansas City, MO 64111, USA. [5]Biomolecular Structure Section, Center for Structural Biology, NCI, NIH, Frederick, MD 21702, USA. [6]Protein Interactions Section, Cancer and Inflammation Program, NCI, NIH, Frederick, MD 21702, USA. [7]Division of Cancer Prevention, NCI, NIH, Bethesda, MD 20892, USA. [8]Neural Development Section, MCGP, NCI, NIH, Frederick, MD 21702, USA. [9]Present address: Centessa Pharmaceuticals, Cambridge, MA 02139, USA. [10]Present address: Ikena Oncology, Cambridge, MA 02210, USA. [11]Present address: Janssen Pharmaceutical Companies, J&J, R&D, Welsh Road McKean Road, Spring House, PA 19477, USA. [12]Present address: Lentigen Technology, Inc. 1201 Clopper Road, Gaithersburg, MD 20878, USA. [13]Present address: Center for Antibody Therapeutics, University of Pittsburgh School of Medicine, Pittsburgh, PA 15261, USA. ✉e-mail: stcroixb@mail.nih.gov

While Matrigel, which is rich in laminin and collagen IV[2], has long been recognized to promote tumor growth[3], the role of collagen I in regulating cancer growth is more complex[4]. Type I collagen (col1), a fibrillar collagen, makes up 25% of total proteins and is the most abundant protein in the human body. Col1 is essential for embryonic viability[5] and accumulates in diseased tissues, for example, during fibrosis and cancer. In early tumorigenesis, epithelial basement membrane, as well as the collagen-rich matrix submucosa, provides a strong barrier to epithelial invasion. Therefore, to become malignant, epithelial cells must acquire the ability to degrade ECM molecules, for example, through upregulation of membrane-bound MT1-MMP[6,7]. Although genetic driver mutations help tumor cells overcome matrix inhibitory cues, cancer cells can still exhibit col1-mediated growth restriction even after transformation. For example, conditional deletion of the *Col1a1* gene in myofibroblasts led to an increase in colon and pancreatic liver metastasis, as well as spontaneous pancreatic tumors[8,9]. Furthermore, colorectal liver metastases are often surrounded by a collagen I rich desmoplastic pseudocapsule that correlates with minimal risk of recurrence, smaller tumors and a better survival[10,11], and such pseudocapsules may represent a teleological hepatic defense mechanism aimed at confining the tumor[12]. Thus, col1 represents an inhibitory restraint that prevents non-transformed epithelial cells from invading collagen rich submucosa[13].

Collagen I can also directly promote tumor growth[14]. For example, collagen-dense breast tumors are correlated with worse patient outcomes[15]. While mechanisms contributing to col1-mediated growth promotion remain ill defined, possibilities include enhanced tumor signaling through increased collagen cross-linking, collagen-mediated immune exclusion, and promotion of metastasis[16,17]. However, using a mutant form of collagen that is resistant to proteolysis, collagen degradation was found to be critical for its ability to promote cancer cell growth in 3D collagen gels and tumor growth in vivo[6,18]. These studies suggest that the growth-promoting activities of collagen I lie downstream of collagen degradation. Recent studies raised the possibility that collagen-derived metabolites such as proline could help fuel the growth of cancer cells, proposing that tumor cells themselves might be able to internalize collagen through non-specific macropinocytosis[19]. Here, we discovered that under microenvironmental stress conditions the tumor-promoting effects of collagen are primarily dependent upon its uptake into TEM8-positive tumor-associated stroma.

Tumor Endothelial Marker 8 (TEM8) is a cell surface protein initially found to be overexpressed in tumor versus normal endothelium of human colorectal cancer[20]. More recent studies have revealed high TEM8 protein expression levels in a variety of solid tumor types where its expression is most prominent in tumor-associated endothelium, pericytes, and CAFs[21]. Soon after its discovery, TEM8 was identified as an anthrax toxin receptor and given the alternative name ANTXR1[22]. Subsequently, a second family member, capillary morphogenesis protein 2 (CMG2), was also identified as an anthrax toxin receptor (ANTXR2) and later shown to be the major receptor for anthrax toxin, presumably due to its higher affinity for protective antigen (PA), the receptor binding subunit of anthrax toxin protein[23,24].

While the role of TEM8 and CMG2 in mediating anthrax toxin delivery is well established, much less is known about their normal physiological functions. Both TEM8 and CMG2 are integrin-like single-pass transmembrane proteins containing an extracellular I- domain (also called a VWA domain) with a metal-ion-dependent adhesion site (MIDAS)[25]. In collagen binding alpha integrins metal ion binding at the MIDAS is critical for collagen binding. Previous studies indicate that TEM8 and CMG2 also bind extracellular matrix (ECM), most notably collagens[26–29] and both TEM8 and CMG2 knockout (KO) mice displayed a gradual buildup of collagens as mice age[30–32]. In humans, loss-of-function mutations in *TEM8* cause Growth retardation, Alopecia,

Pseudoanodontia, and Optic atrophy (GAPO) syndrome, characterized by excess ECM[33].

TEM8 plays an important role in promoting tumor growth—for example, tumors grow slowly in TEM8 global knockout (KO) mice[30,34]. However, it is currently unclear if TEM8s role in collagen homeostasis is related to its tumor-promoting activity. Here we set out to determine if TEM8 is required for internalization of ECM. After elucidating the importance of TEM8 in collagen endocytosis of stromal cells, we then set out to determine the role of collagen in tumor growth and discovered that tumor cells exploit glutamine, a downstream metabolite of collagen degradation, to promote their growth and survival under nutrient-limited conditions, thus uncovering the function of TEM8 in homeostasis and disease.

## Results
### Factors that influence tumor growth in TEM8 wildtype versus knockout mice
TEM8 promotes the growth of late-stage cancers including melanoma, breast, lung, and colon cancer[30,34]. Orthotopic murine breast (EO771), subcutaneous murine glioblastoma (Glioma261), and subcutaneous human pancreatic (HPAC) tumors also showed significantly reduced growth upon host global TEM8 disruption (Fig. 1a–c), providing further evidence that TEM8 in tumor-associated stroma has a broad impact on tumor growth independent of cancer type. To determine if TEM8-mediated growth promotion could be observed in other inbred strains, C57BL6 TEM8 KO mice were backcrossed onto a BALB/C background where similar results were obtained using RENCA kidney, CT26 colon, and 4T1 breast tumor models (Fig. 1d and Supplementary Fig. 1a, b). This TEM8 KO strain, which lacks exon 1, removes the TEM8 signal peptide rendering the coding sequence out-of-frame, and resulting in a null allele[30].

Another previous TEM8 mutant strain was engineered to lack exon 13, encompassing the transmembrane domain, resulting in only the production of a soluble TEM8 (sTEM8) protein detectable in serum[23]. As elevated levels of soluble TEM8 were detected in serum from cancer patients (Supplementary Fig. 2), we examined if sTEM8 protein was important for tumor growth. Both TEM8 KO and sTEM8 mutant strains displayed similar blunted tumor growth when challenged with MC38 murine colon tumors (Fig. 1e), suggesting full-length membrane-bound TEM8 is responsible for the tumor-promoting phenotype. MC38 tumor growth was reduced similarly in TEM8 KO versus WT mice on a syngeneic immunocompetent C57BL/6, immunodeficient NU(NCr)-*Foxn1^{nu}* (athymic nude), or a B6.CB17-*Prkdc* (SCID) background indicating that the pro-tumorigenic function of TEM8 is largely independent of the adaptive immune system (Supplementary Fig. 3).

The cancer cell line-derived solid tumors previously tested in TEM8 WT and KO mice are models of late-stage aggressive disease. We tested its role in early spontaneous tumor formation by generating compound mutant FVB mice containing *TEM8* null alleles and the MMTV-PyMT transgene[35]. MMTV-PyMT mice were previously used to demonstrate a role for COL6 in promoting spontaneous breast tumor formation[36,37] and TEM8 binds COL6[27]. Importantly, this experiment revealed a striking reduction in breast tumor burden in TEM8 KO versus WT animals (Fig. 1f). Although overall levels of Ki67, a marker of proliferation, were not significant reduced, immunofluorescence staining using the collagen binding probe CNA35[38] revealed increased collagen levels in TEM8 KO tumors (Supplementary Fig. 4a, b). Furthermore, tumors developed in 100% of the TEM8^{+/+}/PyMT but only 93% of the TEM8^{−/−}/PyMT strain. We conclude that TEM8 promotes tumor growth from the earliest stages of progression and that TEM8 small molecule antagonists could have potential utility in cancer chemoprevention, an area warranting further investigation.

While the impact of host-derived TEM8 on spontaneous and transplanted tumor growth was observed using multiple tumor

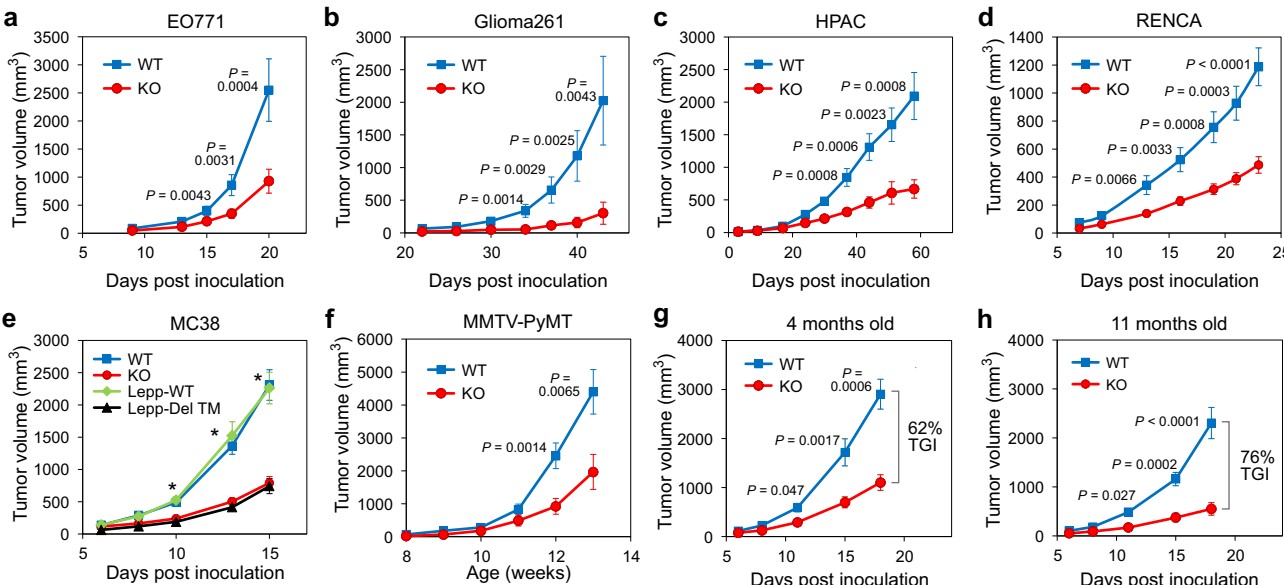

**Fig. 1 | Factors influencing TEM8 dependent tumor growth.** Orthotopic growth of **a** EO771 mammary or subcutaneous growth of **b** Glioma261 tumors in C57BL6 TEM8 WT or KO mice or subcutaneous growth of **c** HPAC pancreatic tumors in athymic nude TEM8 WT or KO mice. $n = 21$ (WT) or 19 (KO) (EO771), 8 (WT) or 15 (KO) (Glioma261), and 10 (WT) or 12 (KO) (HPAC) biologically independent animals per group. **d** Growth of RENCA kidney tumors in BALB/c TEM8 WT or KO mice. $n = 14$ biologically independent animals per group. **e** Growth of MC38 tumors in C57BL6 mice containing a TEM8 knockout (KO) allele, or a deletion of the transmembrane domain (Lepp-Del-TM) along with corresponding wildtype controls. $n = 28$ (TEM8 WT), 24 (TEM8 KO), 28 (Lepp WT), and 25 (Lepp-Del-TM) biologically independent animals per group. *; $P < 0.0001$ between WT and KO and between

Lepp-WT and Lepp-Del-TM at 10-, 13- and 15-days post inoculation. **f** Spontaneous mammary tumor growth in MMTV-PyMT transgenic TEM8 WT and KO on an FVB background. Total tumor burden was measured weekly until mice reached 13 weeks of age. Tumors developed in 100% of TEM8 WT and 93% of TEM8 KO mice. $n = 30$ biologically independent animals per group. Growth of MC38 tumors in TEM8 WT or KO mice at 4 (**g**) or 11 months (**h**) of age. TGI; Tumor growth inhibition. $n = 12$ (4 months) or 15 (11 months) biologically independent animals per group. Data are denoted as mean ± s.e.m. Statistical analysis was calculated using unpaired T tests comparing tumor volume from WT and KO mice (or Lepp-WT and Lepp-Del-TM) at the same day post inoculation. Source data are provided as a Source Data file.

types and genetic backgrounds, the extent of tumor growth delay is variable and appeared to be at least partly model dependent, with some models relying more on host TEM8 than others. Because housing temperature has been found to substantially impact tumor growth[39] we also compared tumor growth rates in mice housed at subthermoneutral (21–22 °C) versus thermoneutral temperature (30–31 °C), but found no alterations in the tumor growth restriction depending on the host TEM8 status (Supplementary Fig. 5). Next, we evaluated the impact of age on the growth of tumors in TEM8 WT and KO that were 4 or 11 months old. While tumor growth was slower in aged mice, as expected from previous studies[40], the difference in tumor growth between TEM8 WT and KO mice also increased with age. In 4-month-old mice TEM8 loss resulted in a 62% tumor growth inhibition (TGI) which increased to 76% by 11 months (Fig. 1g, h). Because collagen levels accumulate gradually in subcutaneous tissues of TEM8 KO mice as they age[30], these results support the possibility that collagen is involved in the tumor suppression phenotype that results from TEM8 deletion in host-derived stromal cells.

## TEM8 + tumor-associated fibroblasts and endothelial cells promote tumor growth

While several cell types in the tumor-associated stroma overexpress TEM8, including endothelial cells, fibroblasts, and pericytes, the specific TEM8 + cell types responsible for supporting tumor growth are hitherto unknown. To address this, we sought to create TEM8 conditional KO mice by crossing our *Tem8* floxed strain to stromal-selective Cre transgenic mice including endothelial *Tie2-cre*[41] or *VE-cad-CreERT2*[42] and the fibroblast *Fsp1-cre*[43,44]. To verify cre expression, first each of the cre deleter lines was crossed with a reporter strain[45] that expresses GFP upon cre-mediated deletion. Co-

immunofluorescence (IF) staining of tumors from the double transgenic mice verified the expression of Cre in tumor-associated vascular cells and CAFs (Fig. 2a, b). While each of the endothelial cre-deleter strains expressed cre in CD31 + tumor endothelial cells, in our tumor models we also detected cre expression in vascular pericytes, although expression in fibroblasts was absent (Supplementary Fig. 6a). Thus, for tumor studies these are better termed vascular deleter strains (i.e., cre is expressed in both endothelial cells and pericytes) rather than pure endothelial deleter strains. Next, each of the cre deleter strains was crossed with the TEM8 conditional knockout to create vascular-or fibroblast-TEM8 deleted KO mice, which were then challenged with various tumors (Fig. 2c–e). These studies revealed that TEM8 expression in VE-cadherin+ or Tie2+ vascular cells plays an important role in the growth of B16 melanoma and MC38 colon tumors (Fig. 2c, d). In contrast, MC38 tumor growth was not affected by TEM8 expression in Fsp1+ fibroblasts (Fig. 2e). On the other hand, TEM8 in Fsp+ fibroblasts promoted growth of SW620 colon tumors, as well as kidney tumors (RENCA) and melanoma (UACC) (Fig. 2e). Surprisingly, TEM8 + vascular cells in the UACC model played a minor role, if any, in tumor growth (Fig. 2d). Based on IF staining, endothelial cells, and pericytes comprised most of the non-hematopoietic stromal cell population of MC38 and B16 tumors, as few CAFs (SMA + cells outside the vasculature) were detected in these tumors (Supplementary Fig. 6b). In contrast, CAFs were the predominant stromal cell type in the RENCA, UACC and SW620 models (Fig. 2b and Supplementary Fig. 6b). These studies indicate that the majority of the TEM8 + stroma promotes tumor growth independent of the lineage from which it derives, with CAFs playing a more prominent role in desmoplastic tumors where their proportion is highest.

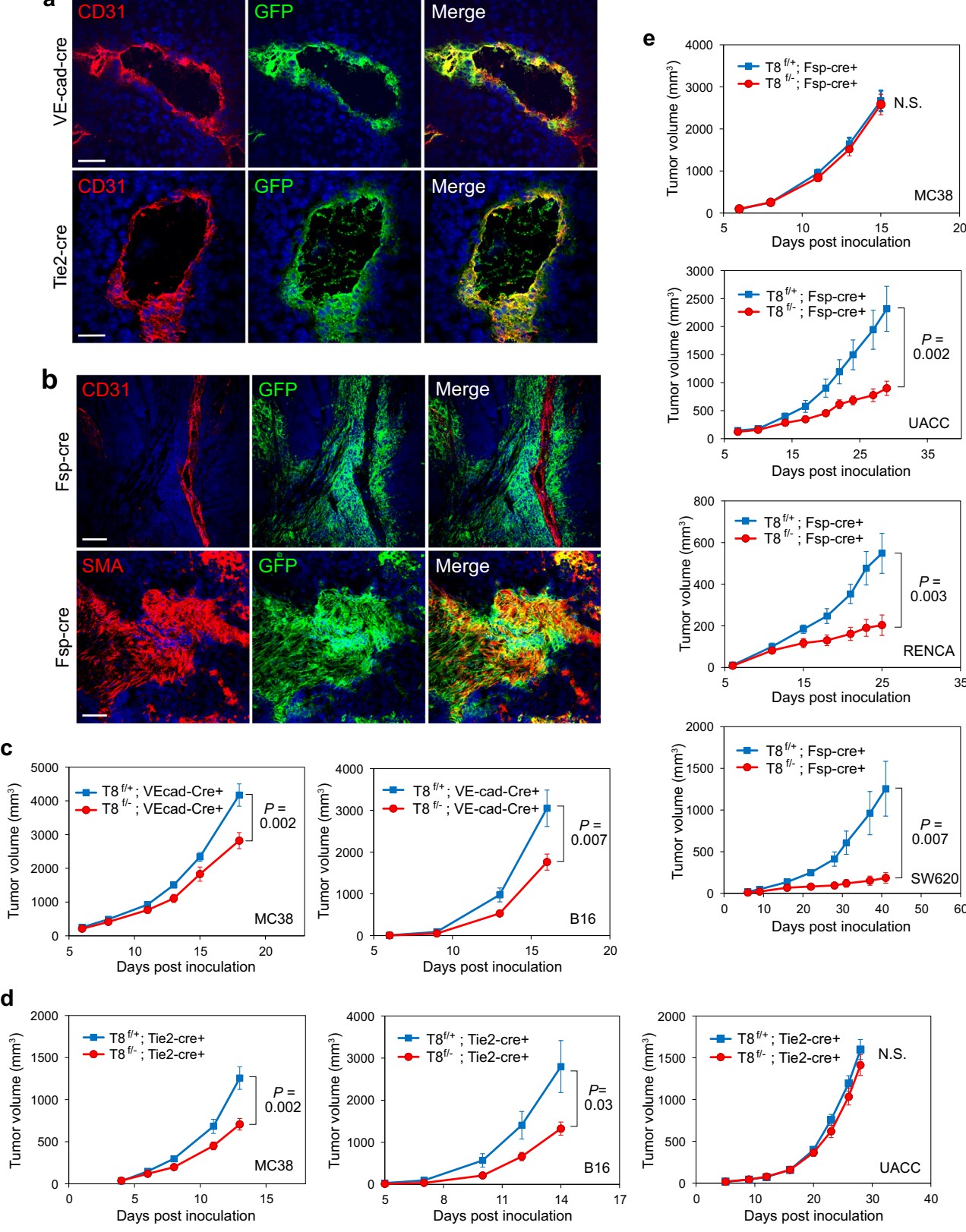

## TEM8 mediates binding and uptake of collagen

Because TEM8 binds collagen and is functionally important in both tumor-associated fibroblasts and vascular endothelial cells and/or pericytes, all of which are known to produce collagen, we hypothesized TEM8-mediated collagen binding and/or uptake is important for tumor growth. To verify if TEM8 could bind collagen, we developed an ELISA using alkaline phosphatase-tagged TEM8 extracellular domains (TEM8-AP) to bind immobilized col1. As a positive control, plates were also coated with the PA subunit of anthrax toxin. Furthermore, because the binding of collagen to the I-domain of integrins is metal ion dependent, we also constructed a TEM8-D150A-AP mutant for comparison, as the negatively charged aspartic acid residue D150

**Fig. 2 | Identification of TEM8 + stromal cells that regulate tumor growth.** Co-immunofluorescence (IF) staining was used to detect CD31 + vasculature (pseudo-colored red) and cre (GFP, green) in **a** B16 tumors from transgenic reporter mice expressing Tie-cre or VE-cad-cre, or **b** SW620 tumors from SCID transgenic reporter mice expressing Fsp-cre. Bar = 50 μm. Images were representative of three experiments (n = 3 animals per group). Subcutaneous tumor growth was monitored in VE-cadherin-cre (**c**), Tie2-cre (**d**) or Fsp-cre (**e**) conditional Tem8 KO strains on a C57BL6 (MC38, B16), BALB/c (RENCA) or SCID (UACC, SW620) background. For MC38, n = 14 (TEM8[flox]+; VE-cad-Cre+), 11 (TEM8[flox/-]; VE-cad- Cre+), 16 (TEM8[flox]+; Tie2-Cre+), 14 (TEM8[flox/-]; Tie2-Cre+), 14 (TEM8[flox]+; Fsp-Cre+), and 15 (TEM8[flox/-]; Fsp-Cre+) biologically independent animals per group. For B16, n = 15 (TEM8[flox]+; VE-cad-Cre+), 19 (TEM8[flox/-]; VE-cad-Cre+), 13 (TEM8[flox]+; Tie2-Cre+) and 13 (TEM8[flox/-];Tie2-Cre+) biologically independent animals per group. For RENCA, n = 14 (TEM8[flox]+; Fsp- Cre+) or 15 (TEM8[flox/-]; Fsp-Cre+) biologically independent animals per group. For UACC, n = 21 (TEM8[flox]+; Tie2-Cre+), 17 (TEM8[flox/-]; Tie2-Cre+), 12 (TEM8[flox]+; Fsp-Cre+) or 15 (TEM8[flox/-]; Fsp-Cre+) biologically independent animals per group. For SW620 n = 5 (TEM8[flox]+; Fsp-Cre+) or 10 (TEM8[flox/-]; Fsp-Cre+) biologically independent animals per group. Data are denoted as mean ± s.e.m. Source data are provided as a Source Data file.

(corresponding to D254 in integrin α2) coordinates metal ion binding to the MIDAS motif (Fig. 3a). WT TEM8-AP, but not TEM8-D150A-AP or AP alone, robustly bound PA and collagen I (Fig. 3b) demonstrating that TEM8 binds collagen through its MIDAS domain. Notably, TEM8 bound col1 and collagen VI, but not laminin or collagen IV (Fig. 3c), confirming collagen selectivity.

To determine if TEM8 on live cells could also bind ECM, next we tested binding of the TEM8 negative CHO-PR230 parental cells (CHO) and TEM8 overexpressing CHO cells (CHO- TEM8)[21] (Fig. 3d and Supplementary Fig. 7a) to various ECM molecules (Fig. 3e). CHO cells were selected for this assay because of their low levels of endogenous collagen receptors[46]. This ECM binding assay revealed strong binding of CHO-TEM8 cells to col1, although a lower level of binding to collagen VI was also detected. We also performed a FITC-labeled collagen I (FITC-Col) internalization assay which involved incubating cells with FITC-Col for 16 h and using flow cytometry to measure FITC transiently trapped inside cells. This assay revealed FITC signal in 75.1% of CHO-TEM8 cells following FITC-col exposure (Supplementary Fig. 7b). However, only 5.3% FITC-col uptake was observed in parent CHO cells confirming low levels of endogenous col1 internalization receptors in the control cells. The same assay was then used to test TEM8 wildtype and knockout Tumor Stromal Cells (TSCs) which were derived from tumors grown in mice harboring TEM8 conditional floxed alleles[21]. After tumor dissociation and isolation of TEM8 positive cells using anti-TEM8 antibody-linked magnetic beads, TSC-WT cells with TEM8 conditional alleles were transiently exposed to Cre in cell culture to create the TEM8 knockout subline called TSC-KO (Supplementary Fig. 7c, d). These studies revealed 55% FITC-positive TSC-WT cells. However, 11.4% of TSC-KO cells also became FITC positive following FITC-col exposure, indicating that, while TEM8 is the predominant receptor responsible for col1 internalization in TSCs, some col1 may also enter TSC-KO cells through a TEM8 independent pathway.

After verifying that TEM8 on live cells can bind col1, next we assessed the impact of TEM8 on col1 uptake and degradation in cells. For these studies, CHO or CHO-TEM8 cells were cultured on a matrix of FITC-Col. Strikingly, we found that the FITC-Col surrounding CHO-TEM8 cells was entirely degraded resulting in cell islands completely devoid of col1, while the FITC-Col surrounding CHO or CHO-CMG2 cells remained largely intact (Fig. 3f and Supplementary Fig. 8a). A similar degradation of FITC-col was observed with TSC-WT cells, but not with TSC-KO cells (Supplementary Fig. 8b). When FITC-Col was layered on top of the cells a similar loss of col1 was observed only in CHO-TEM8 cells (Supplementary Fig. 8c). Collagen internalization into TEM8 + cells could also be visualized by adding soluble FITC-Col to the extracellular medium followed by rinsing (Fig. 3g). To determine if col1 was being transported into a lysosomal compartment, cells were treated with the lysosomal inhibitor NH₄Cl, which resulted in a dramatic intracellular accumulation of FITC-Col (Supplementary Fig. 9a). Internalized FITC-col colocalized with TEM8 and also with LysoTracker Red, a pH sensitive dye, verifying degradation in acidic lysosomes (Supplementary Fig. 9b, c).

To further assess col1 internalization, next Western blotting was used to evaluate the impact of col1 exposure on TEM8 protein levels. This assay revealed a marked decrease in TEM8 protein 24 h following col1 exposure (Fig. 3h). Collagen VI exposure also resulted in a partial reduction in TEM8 levels, while collagen IV, fibronectin, and BSA, did not result in any TEM8 loss.

## Identification of TEM8 mutants that fail to bind collagen

I-domains are present in alpha integrins, where they function to bind collagens and other ECM proteins in a metal ion-dependent manner. To identify residues in the TEM8 protein required for collagen uptake, we compared the three-dimensional crystal structure of the TEM8 I-domain with that of the collagen-binding α-integrins, including the ITGA2 I-domain which was crystalized with a collagen peptide[47]. Because alpha integrins can bind collagen in a metal ion-dependent manner, and mutations in TEM8 and CMG2 that prevent metal ion binding or block PA binding to the MIDAS motif are known, we used this information to select, in addition to D150A, three other conserved candidate MIDAS mutations, D50A, S54A and T118A, that were also likely to disrupt TEM8-collagen binding (Fig. 3a). We also tested two mutations, E152V and H154V, which reside on the surface of TEM8 and are predicted by homology modeling with integrin-α2 to directly block the interaction of TEM8 with collagen fibrils (Fig. 4a, b). E152 and H154 are highly conserved in TEM8 and are also conserved in all four collagen-binding alpha integrins, i.e., α1, α2, α10, and α11, where they have been shown by site-directed mutagenesis and crystal structure studies to interact directly with collagen[47,48]. TEM8-AP fusions with each of the missense mutations revealed these residues indeed were critical for col1 binding (Fig. 4c).

We next created stable cell lines expressing each of the TEM8 mutant receptors, and included a Q137L missense mutation identified in a GAPO patient[49]. Flow cytometry revealed conformationally correct TEM8 expression on the cell surface (Supplementary Fig. 10a). Next, two assays were used to assess the binding and uptake of col1 into mutant cells. The first involved incubating cells on wells precoated with a gel of FITC-Col then monitoring the fluorescence of degraded FITC-Col released into culture supernatants 48 h later. This assay revealed an ~3-fold increase in soluble FITC from CHO-TEM8 versus CHO cells while each of the mutations had soluble FITC levels similar to TEM8 negative cells (Fig. 4d). The second, the FITC-col internalization assay described above, used flow cytometry to measure FITC-col trapped inside cells. This assay revealed a 5-fold increase in col1 levels in CHO-TEM8 cells, while each of the mutants blocked col1 uptake (Fig. 4e and Supplementary 10b). A small increase in uptake was still observed with the S54A and H154V mutants suggesting that these mutants may be less effective than the others at blocking col1 uptake.

## Tumor growth in vivo depends on TEM8-mediated collagen uptake

Having identified point mutations that block col1 uptake, next we sought to explore the role of stromal cell TEM8-collagen binding in tumor growth in vivo by creating a mutant TEM8 knock-in mouse model that was defective in TEM8 mediated col1 binding and uptake. For this, we used CRISPR/Cas9 gene editing to introduce the E150V mutation (corresponding to E152V in human TEM8) into the *TEM8* locus of C57BL/6 mice (*TEM+/E150V*), (Supplementary Fig. 10c, d). Heterozygous mice were then intercrossed to obtain *TEM8E150V/E150V* offspring, which manifested similar phenotypes as *TEM8 KO* mice, such as

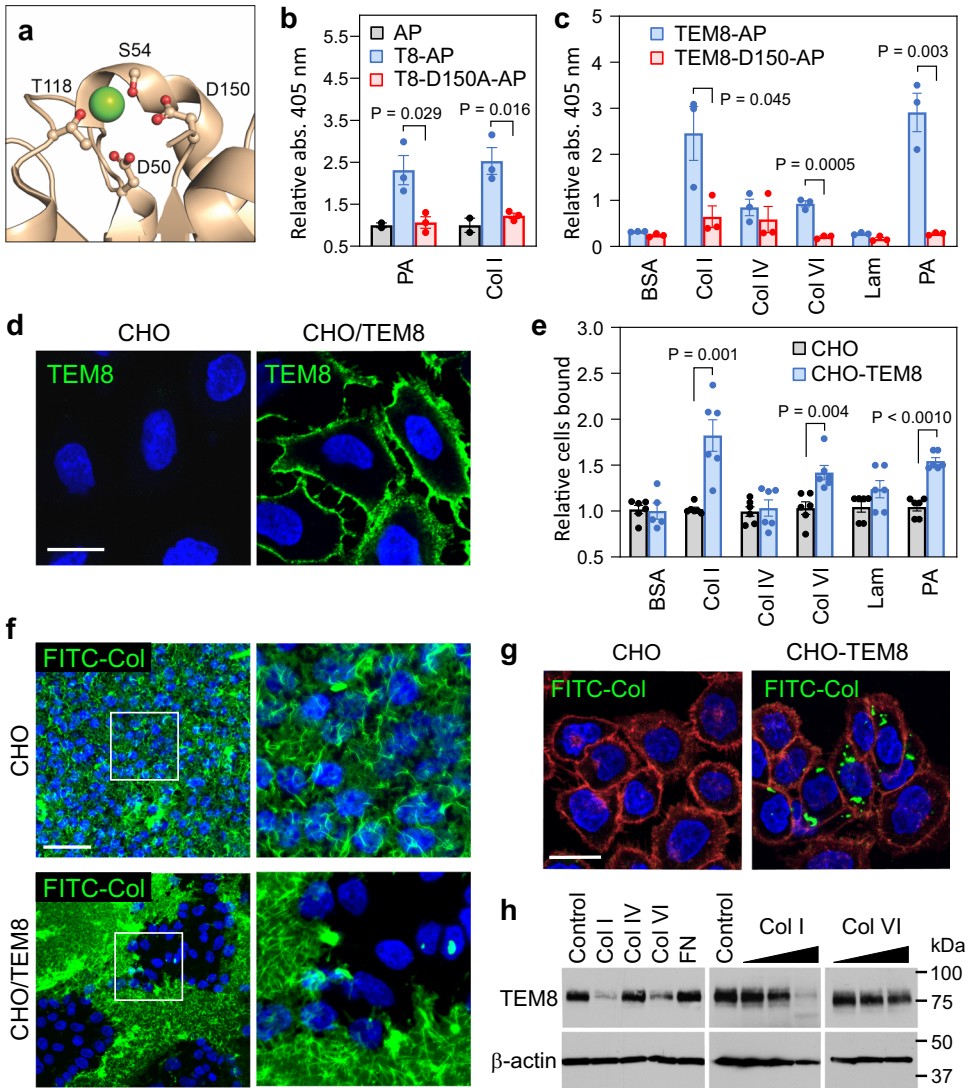

**Fig. 3 | The interaction of TEM8 and collagen I. a** Image depicting amino acids in the TEM8 MIDAS motif that coordinate the metal ion (green). **b** ELISA was used to measure the binding of AP, TEM8-AP, and D150A-AP to col1. PA was included as a positive control. $n = 2$ (AP, negative control) or 3 (TEM8-AP and TEM8-D150-AP) biologically independent samples per group. Statistical comparison between AP and D150A- AP was performed using an unpaired $T$ test. **c** ELISA was used to measure the binding of AP and D150A-AP to various ECM molecules. PA was included as a positive control. $n = 3$ biologically independent samples per group. Statistical comparison between AP and D150A-AP was performed using an unpaired $T$ test. **d** IF staining was used to detect TEM8 (green) in CHO and CHO-TEM8 cells. Bar = 20 μm. Images were representative of three experiments. **e** A cell binding assay was used to measure binding of CHO-TEM8 cells to various ECM molecules.

$n = 6$ biologically independent samples per group. Statistical comparisons between CHO and CHO-TEM8 were performed using an unpaired $T$ test. **f** IF staining was used to detect CHO-mediated degradation of an underlying FITC-col gel (green). Cell nuclei were visualized using DAPI (blue). Bar = 50 μm. Images were representative of three experiments. **g** IF staining was used to detect collagen uptake after adding soluble FITC collagen (green) to the media. CellMask orange was used to visualize cell membranes (red) and Hoechst 33342 to visualize nuclei (blue). Bar = 20 μm. Images were representative of three experiments. **h** Western blotting was used to detect changes in TEM8 expression following exposure of CHO-TEM8 cells to various ECM molecules. Wedge: Col1; 10, 25, and 50 μg/mL, ColVI; 1, 10, 25 μg/mL. Images were representative of three experiments. Data in **b**, **c**, and **e**, are denoted as mean ± s.e.m. Source data are provided as a Source Data file.

dwarfism, a bossed forehead, and misaligned incisors (Supplementary Fig. 11). $TEM8^{+/+}$ and $TEM8^{E150V/E150V}$ littermates were then challenged with MC38 colon tumors. Tumor growth was severely blunted in the $TEM8^{E150V/E150V}$ knock-in compared to wildtype mice, with blunted tumor growth similar to that observed in TEM8 knockout versus wildtype mice (Fig. 4f). These results support the idea that TEM8 binding to collagen in tumor-associated stroma is responsible for the tumor-promotion phenotype, as the E150V mutant, which is unable to bind col1, phenocopies the global TEM8 global knockout.

**TEM8 expression is regulated by environmental stress**
We next examined the environmental triggers that lead to elevated TEM8 expression in tumors. Previous studies demonstrated *TEM8*

mRNA is upregulated up to 20-fold in ischemic hind limb muscle tissue following femoral artery ligation[50]. TEM8 protein levels were also found to be upregulated fivefold in cultured human microvascular endothelial cells (HMECs) in response to growth factor deprivation[34]. Like normal tissues that become ischemic, in solid tumors, the high metabolic demand of cancer cells depletes nutrients locally resulting in chronic stress and ischemic-like conditions. While hypoxia had no impact on TEM8 expression, consistent with earlier studies, we confirmed that growth factor deprivation caused an increase in TEM8, but not CMG2, in two primary endothelial cell models−HMEC and TIME (Fig. 5a, b). A dramatic increase in TEM8 in TSCs isolated from tumors[21] was also observed following serum deprivation (Fig. 5c). Importantly, while proline levels in the media did not regulate TEM8 expression,

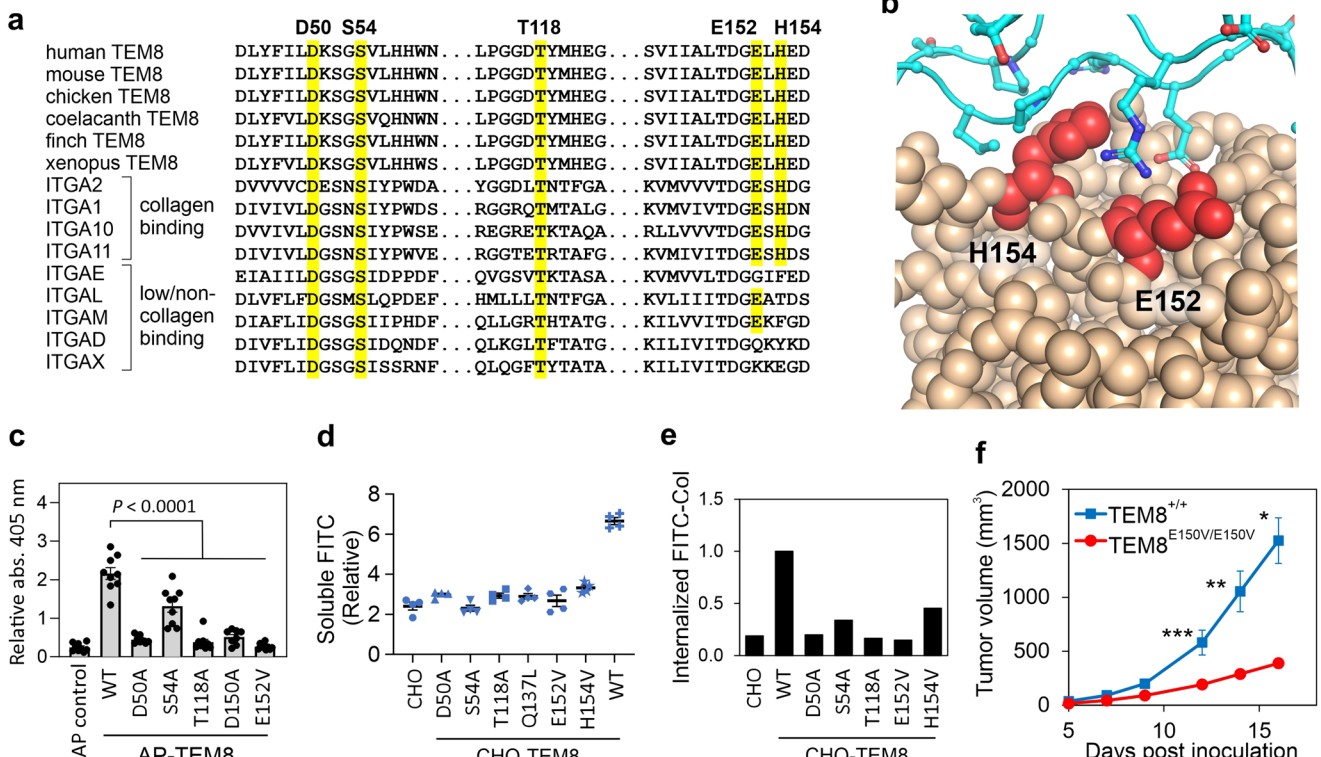

**Fig. 4 | Mutations in TEM8 block collagen binding and tumor growth. a** Alignment of TEM8 with the known collagen-binding site of integrins. The conserved mutations in this study, including the Glutamine (E) and Histidine (H) found in collagen binding integrins are highlighted (yellow). **b** Image depicting the conserved TEM8 surface residues, E152 and H154, predicted to contact collagen based on homology with integrin alpha 2. **c** An ELISA was used to measure the binding of AP, AP-TEM8 (WT) and various AP-TEM8- mutants to col1. $n \geq 8$ biologically independent samples per group. Statistical analysis was calculated using one-way analysis of variance with a Tukey's test. **d** A FITC release assay was used to measure soluble FITC in the supernatant of CHO cells, CHO cells expressing wildtype TEM8 (WT) or various TEM8 mutants following culture in a FITC-Col gel. $n = 4$ biologically

independent samples per group. $P < 0.0001$ between CHO- TEM8 WT and each of the CHO-TEM8 mutants. Statistical analysis was calculated by using one-way analysis of variance with a Tukey's test. **e** Flow cytometry was used to measure FITC in CHO, CHO-TEM8 (WT), and CHO-TEM8 mutant cells following FITC-Col treatment. **f** Growth of MC38 in TEM8$^{+/+}$ or TEM8$^{E150V/E150V}$ mice. $n = 20$ (TEM8$^{+/+}$) and 14 (TEM8$^{E150V/E150V}$) biologically independent animals per group. Mice used in this study were 4 to 5 months old. Statistical analysis was calculated using unpaired $T$ tests comparing tumor volume from WT and KI mice on the same day post inoculation. *; $P < 0.0001$, **; $P = 0.002$, ***; $P = 0.009$. Data are denoted as mean ± s.e.m. Source data are provided as a Source Data file.

glutamine deprivation caused a time-dependent increase in TEM8 protein in both TSCs and HMECs (Fig. 5d, e). TEM8 expression levels in vivo were found to be highest in stromal cells adjacent to tumor cells (Fig. 5f, g) suggesting local stress caused by tumor-mediated nutrient depletion may regulate TEM8 expression in stromal cells in vivo. Taken together these results suggest that nutrient deprivation and other potential stressors in the tumor microenvironment likely drive TEM8 overexpression in the tumor-associated stroma.

**Collagen provides a source of glutamine for tumor cells under stress conditions**

Elevation of TEM8 expression in stroma in response to nutrient deprivation suggests that TEM8 may be part of a stress-induced response pathway that helps cells survive under nutrient-limited conditions. Collagen, the most abundant protein in humans, may act as an alternative fuel source for TEM8 + cells in a nutrient-deprived environment. Collagen is comprised of about 23% proline, a unique amino acid which can be metabolized to produce ATP under stress[51]. Proline can be converted into pyrroline-5-carboxylate (P5C), glutamate (Glu), and then α-ketoglutarate (α-KG), which can be used to fuel the TCA cycle. In stromal cells, which express prolidase (PEPD), proline dehydrogenase (PRODH), P5C dehydrogenase, and glutamine synthetase (GLUL), collagen can also be rapidly converted into glutamine which can potentially act as a vital source of energy for nutrient-deprived cells (Fig. 6a). As CHO cells lack enzymes needed to produce proline

from ornithine or glutamine, they require exogenous proline supplementation for survival[52]. CHO cells also depend on glutamine and FBS. However, when cultured under serum-limited conditions without glutamine or proline, overexpression of TEM8 in CHO cells partially rescued cell viability on col1-coated plates, demonstrating that TEM8 may allow subsistence under nutrient-limited conditions (Supplementary Fig. 12a).

To further understand the importance of collagen and metabolites in stromal cell viability, next we used TEM8 wildtype and knockout TSCs[21]. TSCs showed dependence on exogenous serum and glutamine, but not proline. When stressed by culturing for 48 h with 0.5% FBS and no glutamine and proline, both TSC-WT and TSC-KO cells displayed ~70% reduction in viability (Fig. 6b). Addition of exogenous col1 partially rescued viability of TEM8 + TSCs but not TEM8 null TSCs.

To determine if TEM8 and collagen directly affect tumor cells, next we exposed SW620 colon, PC9 lung, and HPAC pancreatic cancer cells to nutrient stress and compared viability in the presence or absence of col1. Because some tumor cells express a low but detectable level of TEM8 (Supplementary Fig. 12b), we also compared the viability of HPAC with HPAC- T8-KO cells[21]. Not surprisingly, each of the tested cancer cell lines displayed 30–60% reduced viability when cultured in the presence 0.5% versus 10% dialyzed serum, demonstrating partial dependence on exogenous growth factors (Fig. 6c and Supplementary Fig. 12c, d). While no further loss of viability was observed in response to proline depletion, a prominent reduction in viability in response to

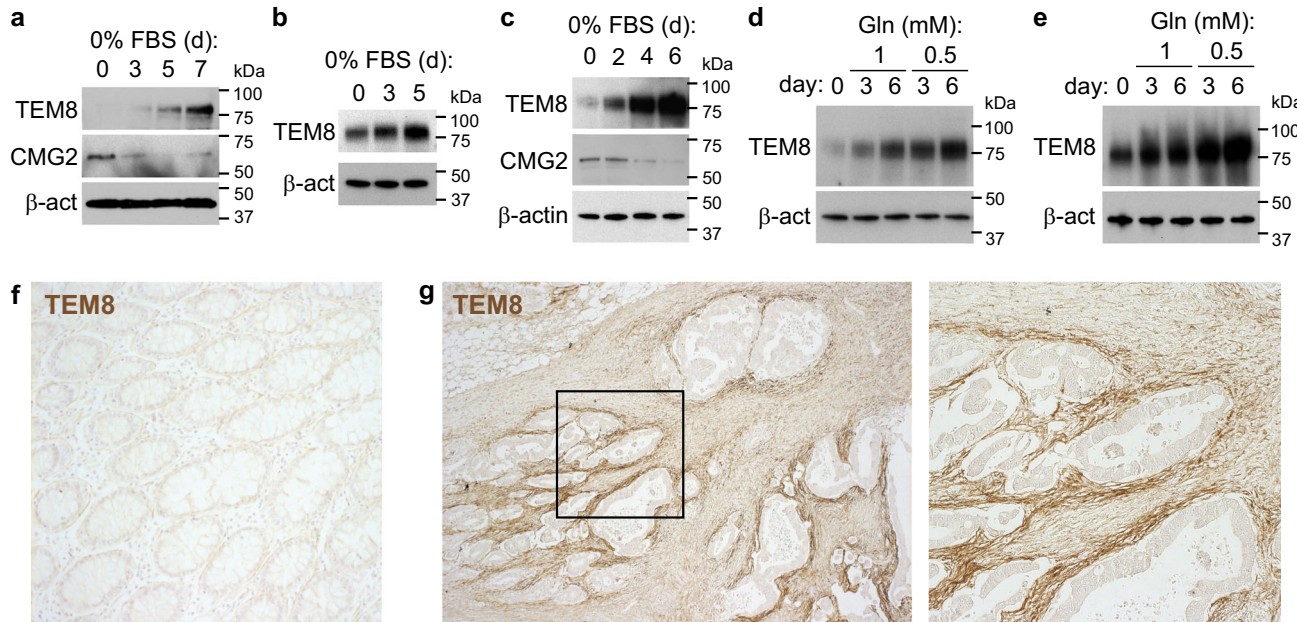

**Fig. 5 | TEM8 expression is regulated by microenvironmental stress.** Western blotting was used to detect TEM8 or CMG2 protein expression in HMECs (**a, d**), TIME (**b**), or TSCs (**c, e**) following serum deprivation (**a–c**) or glutamine (Gln) deprivation (**d, e**). β-actin (β-act) was used as a loading control. Images were representative of three experiments. IHC was used to assess TEM8 protein expression in normal adjacent human colon (**f**) or colorectal cancer (**g**). Bar = 100 μm (**f**) or 200 μm (**g**). Images were representative of three experiments. Source data are provided as a Source Data file.

glutamine withdrawal was consistently detected, with stressed cancer cells displaying only ~29% (SW620), 6% (PC9), or 23% (HPAC) viability compared to controls (Fig. 6c and Supplementary Fig. 12c, d). Interestingly, under these stress conditions addition of exogenous col1 to cancer cells further reduced cell viability to 6% (SW620), 2.2% (PC9), and 12% (HPAC), although cancer cell TEM8 status did not impact these responses. Amino acid spiking experiments revealed that glutamine was the only amino acid absent from the nutrient-depleted basal culture media that was able to rescue viability of each of the cancer cell lines (Supplementary Fig. 12e). Furthermore, supplementation of growth media with glutamine resulted in a significant increase in ATP production in cancer cells, even under nutrient starvation (Fig. 6d and Supplementary Fig. 12f).

Next, we assessed the impact of TSCs on tumor cell survival in response to glutamine deprivation by co-culturing cells for 48 h in the presence of col1. In these assays only tumor cells are luciferase labeled, allowing their viability to be selectively monitored in the presence of TSCs. Remarkably, in the absence of exogenous glutamine, cancer cell viability on col1 plates could be completely rescued by co-culturing with TEM8 WT TSCs, but not TEM8 KO TSCs (Fig. 6e and Supplementary Fig. 13a–c). However, supplementation of the TSC-KO/cancer cell co-cultures with glutamine completely restored cancer cell viability.

To verify if collagen is being converted to glutamine in TSCs, we knocked down prolidase, which catalyzes the final and rate-limiting step in the degradation of collagen, releasing free proline, and also knocked down proline dehydrogenase and glutamine synthase, key enzymes involved in the conversion of proline to glutamine, by using two independent shRNAs per gene (Fig. 6f–h). Knockdown of each of these genes in WT TSCs blocked their ability to rescue tumor cell viability on col1 plates under nutrient deprivation while supplementation with glutamine, but not proline, rescued viability (Fig. 6i–k and Supplementary Fig. 13d, e). Next, we used mass spectrometry to evaluate the level of all 20 amino acids in the supernatant of TSC-WT or TSC-KO cells and discovered that glutamine was the only amino acid drastically depleted from TSC-KO cultures (Fig. 6l). To exclude the possibility of rescue from soluble factors and confirm that the rescue is

through TSC-secreted glutamine, we added to nutrient-starved co-cultures L-asparaginase, which hydrolyzes extracellular glutamine to glutamate and ammonia, thereby blocking uptake of TSC secreted glutamine by the cancer cells. L-asparaginase blocked TSC mediated rescue of each cancer cell line tested—SW620, PC9, and TEM8-KO HPAC (Fig. 6m and Supplementary Fig. 13f, g). Taken together these results indicate that the collagen degradation pathway is operative in TEM8 + TSCs, and that the final product, glutamine, a nitrogen-rich amino acid known for its ability to transport between cells, is a key mediator exploited by tumor cells for their survival.

### TEM8 antibodies block collagen uptake, tumor growth, and metastasis

To further explore the impact of TEM8-collagen binding on tumor growth, next, we used in vitro yeast antibody display to develop a fully human TEM8 antibody, called m830, that could block TEM8-mediated col1 internalization. This approach allows the selection of cross-species reactive antibodies, i.e., antibodies that can bind both mouse and human TEM8, avoiding immune tolerance mechanisms. m830 recognized native human and mouse TEM8 with similar high affinity but not human or mouse CMG2 (Supplementary Fig. 14a, b). Importantly, m830 blocked binding of col1 to TEM8 and uptake into both CHO-TEM8 cells and TSCs (Fig. 7a, b and Supplementary Figs. 7d and 14c). Furthermore, m830 prevented TSC-WT cells from releasing glutamine into the supernatant (Fig. 6l).

Next, we tested m830 col1 blocking anti-TEM8 antibodies for tumoricidal activity in vivo. m830 was able to block the growth of multiple tumor types, mirroring tumor growth delays observed in TEM8 KO mice (Fig.7c and Supplementary Fig. 14d). Treatment with 15 mg/kg of m830 (3 × per week for 3 weeks) was well tolerated and had no discernable impact on body weight (Fig. 7d). Furthermore, m830 serum levels following intravenous injection were substantially elevated in TEM8 KO versus WT mice (Supplementary Fig. 14e), consistent with target mediated deposition and highlighting the specificity of m830 for TEM8 in vivo. Based on its specificity and high affinity for TEM8 ($K_D$:1.6 to 3.7 nM for hTEM8 and mTEM8, respectively), ability to block col1, high production yields (>100 mg/L), potent anti-tumor

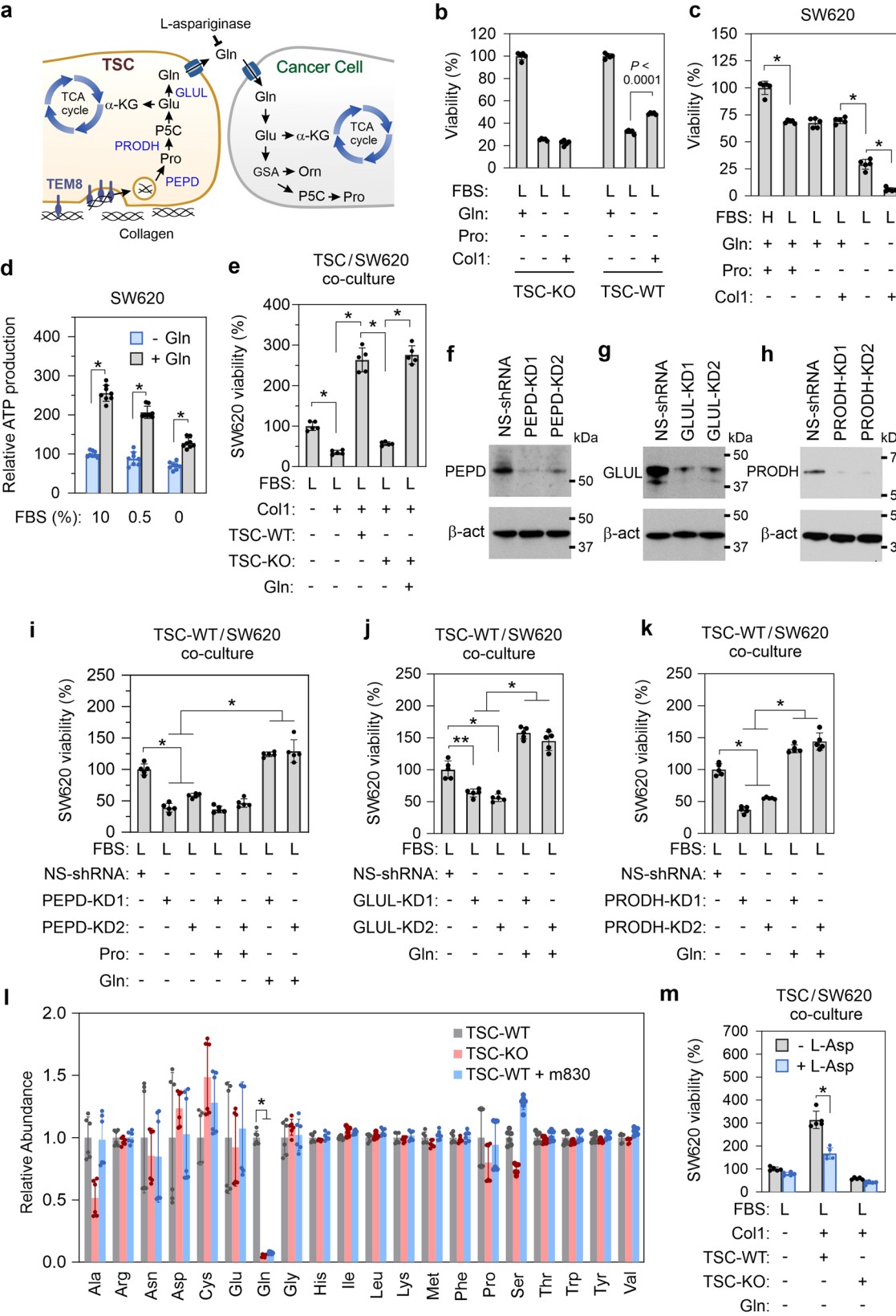

activity and lack of toxicity, the m830 antibody has become our lead fully-human IgG for translation to clinical development.

The inability to control established metastases is the most frequent cause of treatment failure and cancer-related deaths. To determine if col1-blocking anti-TEM8 antibodies could treat disseminated metastases we employed two colon cancer liver metastasis models, as this cancer type frequently metastasizes to the liver where it evokes high TEM8 and col1 levels in stromal infiltrates (Supplementary Fig. 15). MC38 murine or luciferase-tagged HCT-116 human colon tumor cells were seeded into the liver by intrasplenic injection and, one day later, treated with vehicle or m830 anti-TEM8 antibodies (Fig. 7e–h and Supplementary Fig. 16a).m830 significantly reduced liver tumor

**Fig. 6 | TSC-derived glutamine protects cancer cells from microenvironmental stress. a** Depiction of the collagen degradation pathway in TSCs that results in the production of glutamine (Gln), a transportable metabolite that can be exploited by cancer cells. **b** Viability of TEM8 wildtype (WT) or knockout (KO) TSCs under nutrient starvation in the presence or absence of col1. $n = 5$ biologically independent samples per group. Statistical analysis was calculated using an unpaired $T$ test. **c** Viability of SW620 cancer cells under nutrient starvation in the presence or absence of col1. $n = 5$ biologically independent samples per group. *; $p < 0.0001$. Statistical analysis was calculated by using one-way analysis of variance with a Tukey's test. **d** Relative ATP production in SW620 cells in the presence or absence of glutamine (Gln). $n = 8$ biologically independent samples per group. *; $p < 0.0001$. Statistical analysis was calculated by using an unpaired $T$ test. **e** Viability of SW620 cancer cells under nutrient starvation in co-cultures with TEM8 wildtype or knockout TSCs. $n = 5$ biologically independent samples per group. *; $p < 0.0001$. Statistical analysis was calculated by using one-way analysis of variance with a Tukey's test. Western blotting was used to detect PEPD (**f**), GLUL (**g**) or PRODH (**h**)

expression in wildtype TSCs following knockdown (KD) with non-specific control shRNA or gene-specific shRNA. Images were representative of three experiments. Viability of SW620 cancer cells under nutrient starvation upon co-culture with PEPD (**i**), GLUL (**j**), or PRODH (**k**) knock down (KD) TSCs. NS-shRNA: non-specific control shRNA. $n = 5$ biologically independent samples per group. *; $p < 0.0001$, **; $p < 0.0003$. Statistical analysis was calculated by using one-way analysis of variance with a Tukey's test. **l** Relative abundance of each of the 20 amino acids in the supernatants of TSC-WT, TSC-KO, or TSC-WT cells following m830 anti-TEM8 antibody treatment. To minimize experimental variability, data from two experiments were combined ($n = 3$ per group). *; $p < 0.0001$. Statistical analysis was calculated by using one-way analysis of variance with a Tukey's test. **m** Viability of nutrient-starved SW620 cancer cells in TSC co-cultures following treatment with L-Asparaginase (L-Asp). $n = 5$ biologically independent samples per group. *; $p < 0.0001$. Statistical analysis was calculated by using an unpaired $T$ test. Gln; glutamine, Pro; proline, H; high serum, 10% FBS, L; low serum 0.5% FBS. Data are denoted as mean ± s.d. Source data are provided as a Source Data file.

burden and improved overall survival in both models, suggesting that TEM8 antibodies may aid in the treatment of established metastatic disease.

## Improving the efficacy of TEM8 antibody-based therapies

Most current anticancer therapies target either rapidly dividing (i.e., nutrient replete) tumor cells, or VEGF-responsive endothelial cells of the tumor neovasculature—i.e., anti-angiogenic therapy. Because the TEM8/collagen pathway is most important for nutrient-deprived tumor cells, we hypothesized TEM8 blockade could target tumor cell populations missed by conventional therapies. To test this idea, m830 treatment was tested alone or in combination with the conventional chemotherapeutic agents irinotecan and paclitaxel, or bevacizumab, an anti-VEGF antibody. m830 augmented the agents-killing activity in both DLD1 colon and NCI-H460 lung cancer xenografts without added toxicity (Fig. 7i, j and Supplementary Fig. 16b–d). m830 was also tested in combination with anti-PD-1 antibodies (clone RMPI), an immune checkpoint inhibitor that helps T-cells to recognize neoantigens on tumor cells in mice bearing MC38 colon tumors. Anti-TEM8 antibodies improved efficacy of PD-1 blockade in these mice (Fig. 7k). Thus, m830 complements the activity of a diverse group of therapeutic agents including chemotherapy, anti-angiogenic therapy, and immunotherapy, consistent with its ability to target a unique metabolic pathway not directly targeted by conventional cancer therapies.

## Discussion

Previous studies revealed that TEM8 mRNA and protein are expressed at highest levels in the tumor-associated stroma[20,21,27,53]. Although tumor cells frequently express a low level of TEM8, tumors derived from TEM8 WT or KO cancer cells displayed similar growth kinetics in vivo[21]. In contrast, here, we show that the high TEM8 levels in tumor-associated stroma are responsible for the prominent tumor growth-promoting activity of TEM8 in vivo. Solid tumors have variable numbers of stromal cells, with CAFs or myofibroblasts comprising the predominant population in desmoplastic tumors and tumor-associated endothelial cells and pericytes in non-desmoplastic tumors. Each of these cell types can express TEM8 and each of them produces collagen type I and were found to be important for TEM8-mediated tumor growth promotion.

TEM8-mediated growth promotion was found to depend on collagen I uptake which provided TSCs with a vital source of energy under nutrient-limited conditions. Glucose, glutamine, and fatty acids from the circulation provide a primary source of energy for most cells, but access to circulating nutrients is often limited in solid tumors due to their high metabolic demand combined with an inefficient neovasculature. For most solid tumors, including lung, breast, brain, renal and colon cancer, the vascular density within the tumor is lower than that in surrounding normal tissues[54]. Furthermore, the blood vessels present in tumors are often highly dysfunctional—for example, tortuous,

blunt-ended, and incompletely covered with pericytes, which can lead to erratic and even static blood flow[55]. Deficiencies in tumor blood vasculature, high interstitial pressure, and high metabolic demand can result in a local depletion of nutrients and growth factors, creating environmental stress. Most conventional anticancer agents preferentially target rapidly growing tumor cells, i.e., those closest to perfused vessels, and may therefore augment the activity of TEM8 blocking therapies which target the nutrient-stressed compartment.

CAFs, one of the major TEM8 + cell types in the tumor-associated stroma, have been shown to have both tumor-promoting and tumor-restricting activity depending on the context[56,57]. Similarly, stromal-derived collagen I has also been shown to exert both tumor-promoting and tumor-restricting phenotypes[8,9,58]. While mechanisms underlying these seemingly paradoxical outcomes have remained unclear, we posit that solid tumors in vivo can simultaneously experience both collagen-promoting and collagen-restricting forces. In this model, depending on which of these opposing forces predominates, the net balance determines whether collagen I is pro-tumorigenic or anti-tumorigenic. Surprisingly, we found that collagen I could directly block the growth of cancer cells cultured under nutrient stress conditions in the absence of TEM8 + TSCs, the extent of which depended on the tumor model and likely relates to the set of driver mutations facilitating tumorigenesis. On the other hand, collagen I taken up by TEM8 + TSCs provided tumor cells an essential source of glutamine that enabled them to grow when the nutrient supply was poor.

Glutamine is a non-essential amino acid that can become conditionally essential under catabolic stress conditions[59]. Glutamine plays a major role in replenishing TCA cycle intermediates for ATP production and serves as an important carbon and nitrogen donor for the synthesis of proteins, lipids, and nucleotides[60]. The long-recognized glutamine sensitivity of cancer cells has led to a substantial interest in developing therapies that exploit this vulnerability including glutamine analogs, glutamine synthetase inhibitors and glutamine transport blockers[61]. However, clinical translation of these treatments has been limited due to off-target toxicities, as systemic reductions in glutamine can be detrimental[62]. TEM8 targeting represents a localized alternative which may circumvent these problems.

Consistent with our data, studies by Yang et al. showed that CAFs derived from ovarian cancer can secrete high levels of glutamine, while normal fibroblasts could not[63]. Over 60% of intracellular glutamate in cancer cells was derived from CAF-secreted glutamine. While CAF deletion therapy has been considered, concerns about the potential removal of tumor suppressive signals have dampened enthusiasm for this approach[56,64]. Here, by using TEM8 neutralizing antibodies to block tumor growth, we provide an alternative strategy that preserves the inhibitory action of extracellular collagen produced by TSCs while simultaneously preventing its conversion to tumor-promoting metabolites. Therefore, TEM8 antagonists have the potential to deliver a double blow to solid tumors, disabling their exploitation of stromal-

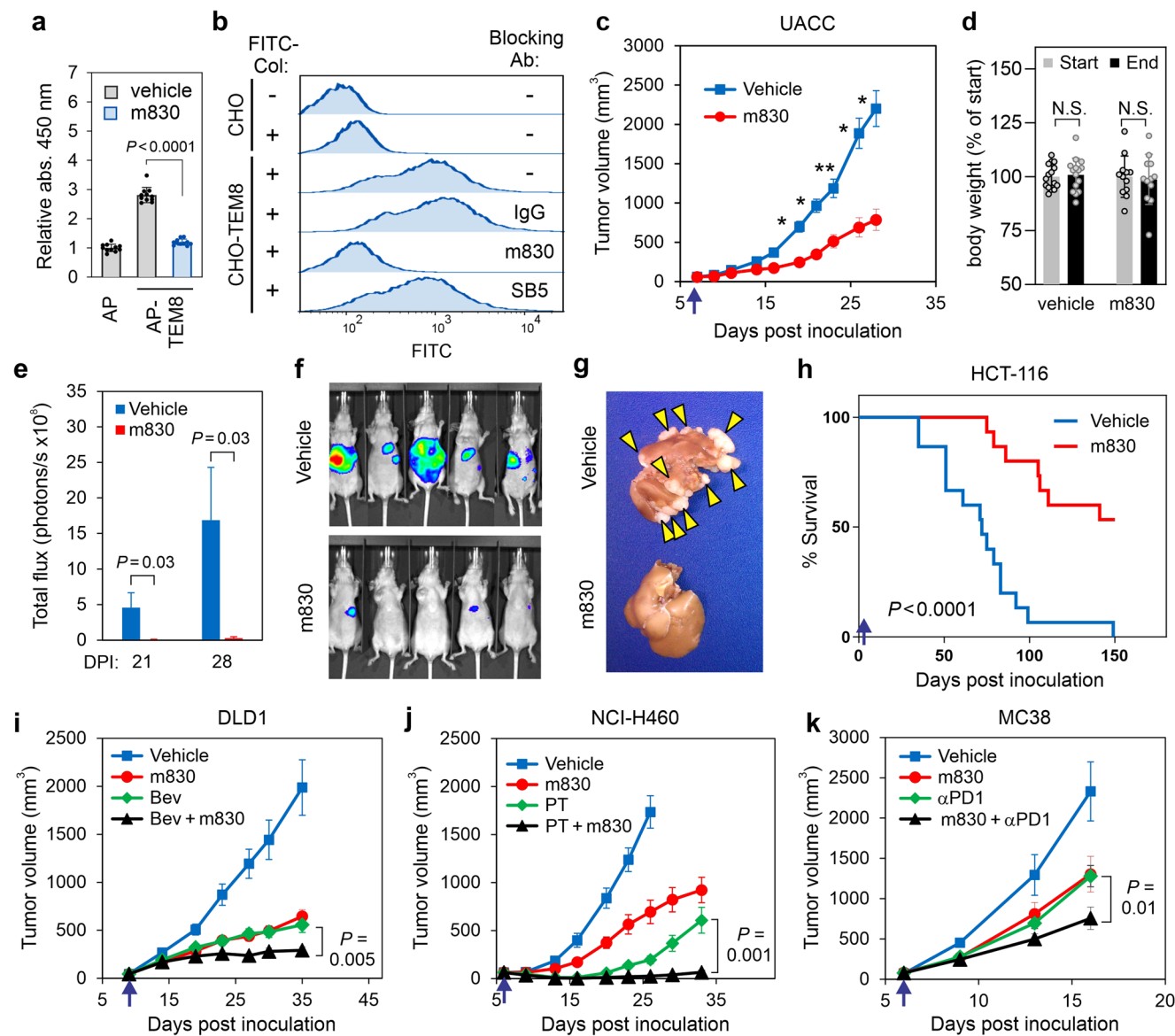

**Fig. 7 | TEM8-collagen neutralizing antibodies block tumor growth. a** ELISA was used to measure the binding of AP (control) and TEM8-AP to col1 in the absence or presence of 20 μg/mL of m830 anti-TEM8 antibodies. $n = 11$ biologically independent samples per group. Statistical analysis was calculated by using an unpaired $T$ test. **b** Flow cytometry was used to quantify FITC-Col uptake in CHO and CHO-TEM8 cells that were treated with non-specific control IgG (IgG) or TEM8 antibodies m830 or SB5. The SB5 anit-TEM8 antibody[27] is unable to block collagen binding and was used as an additional negative control. **c** Growth of UACC melanoma tumors following treatment with 15 mg/kg of m830 antibody. Treatments were administered 3× per week and initiated (arrow) when tumors reached a size of 60 mm³. $n = 18$ (vehicle) or 12 (m830) biologically independent animals per group. *; $P ≤ 0.0001$, **; $P = 0.0002$. Statistical analysis was calculated using unpaired $T$ tests comparing tumor volume from vehicle and m830 treated mice on the same day post inoculation. **d** Average body weights of mice in **c** at treatment start (day 6) and study end (day 28). N.S.: non-significant. $n = 15$ (vehicle) or 12 (m830) independent samples per group. Statistical analysis was calculated by using an unpaired $T$ test. Liver metastases following intrasplenic injection of HCT-116-luc colon cancer was quantified at 21- and 28-days post inoculation (DPI) (**e**) using BLI. Statistical analysis was calculated by using an unpaired T test. Images from five representative mice/group are shown in **f** and examples of excised livers with tumor lesions (arrowheads) are shown in **g**.

Kaplan- Meier survival analysis (**h**). In this study, treatments began with 15 mg/kg one day following tumor cell inoculation (arrow) followed by 5 mg/kg 3 times/ week for 4 weeks. Log-rank analysis: $P < 0.0001$, m830 versus vehicle. $n = 15$/ group. **i** Growth of DLD1 colon tumors following treatment with 15 mg/kg of m830 3× per week, 5 mg/kg bevacizumab (Bev) 2× per week, or a combination of m830 and Bev. Treatments were initiated (arrow) when tumors reached an average size of 50 mm³. $n = 18$ (vehicle), 13 (m830), 13 (Bev), or 14 (m830 + Bev) biologically independent animals per group. Statistical analysis was calculated by using an unpaired $T$ test. **j** Growth of NCI-H460 lung tumors following treatment with 15 mg/kg of m830 3x per week, 30 mg/kg paclitaxel (PT) (qod × 5), or a combination of m830 and PT. Treatments were initiated (arrow) when tumors reached an average size of 60 mm³. $n = 10$ biologically independent animals per group. Statistical analysis was calculated by using an unpaired $T$ test. **k** Growth of MC38 colon tumors following treatment with 10 mg/kg of m830, 3.5 mg/kg anti-PD1 antibody (αPD1, clone RMPI), or a combination of m830 and αPD1. Treatments were administered 3× per week and were initiated (arrow) when tumors reached an average size of 80 mm³. $p = 0.01$ with respect to αPD1 alone. $n = 12$ (vehicle), 10 (m830), 14 (anti-PD1) or 14 (m830 + anti-PD1) biologically independent animals per group. Statistical analysis was calculated by using an unpaired $T$ test. Data in **a** and **d** are denoted as mean ± s.d. Data in **c, i**–**k** are denoted as mean ± s.e.m. Source data are provided as a Source Data file.

derived nutrients while preserving the restriction-mediated functions of intact collagen I. In combination with other anti-cancer agents, TEM8 antagonists provide a promising approach for improved cancer therapy against multiple solid tumor types.

## Methods

All research contained in this study complies with all relevant ethical regulations.

### Vectors

hTEM8/pcDNA3.1 vector encoding human TEM8 (GenBank AF279145.2) was used to synthesize each of the mutations described here using overlap extension PCR. TEM8 mutations were subcloned into AP-TEM8 vectors[27] and sequence verified.

### Cell lines

Cell lines 293, HPAC, SW620, CT26, 4T1, and BALB/3T3 were obtained from the American Type Culture Collection (ATCC, cat nos. CRL-1573, CRL-2119, CRL-2638, CRL- 2539 and CCL-163, respectively). E0771 cells were obtained from CH3 BioSystems (cat. no. 940001) and PC9 cells were from Sigma (cat. no. 90071810). B16, glioma 261, and UACC-64 (UACC) cell lines were from the DCTD Tumor Repository at NCI (Frederick, MD). MC38, RENCA, and CHO-PR230 (CHO) cell lines were gifts of Jeffrey Schlom (NCI, NIH), Jonathan M. Weiss (NCI, NIH), and Stephen H. Leppla (National Institute of Allergy and Infectious Diseases [NIAID]), respectively. CHO-TEM8 and CHO-CMG2 cells were previously described[21], as were HCT-116-luc cells[65].

### Clinical samples

Anonymized human serum samples were obtained from the Research Donor Program (RDP) at the Frederick National Laboratory, the Cooperative Human Tissue Network (CHTN), or were a kind gift from Dr. Oliver Bathe (University of Calgary) with approval from the NIH Office of Human Subject Research (OHSR). Anonymized human FFPE normal or cancer tissue samples were obtained from the Cooperative Human Tissue Network (CHTN) with approval from the NIH OHSR. All clinical protocols were approved by institution-specific investigational review boards, with appropriate patient informed consent.

### Animals

All experiments involving animals were conducted in accordance with protocols approved by the NCI Animal Care and Use Committee. All mice were bred and maintained in a pathogen-free facility certified by the Association for Assessment and Accreditation of Laboratory Animal Care International. Mice were fed Charles Rivers Rat and Mouse 18% autoclavable diet (Cat # 5L79, LabDiet) ad libitum and maintained under conventional housing. TEM8 KO mice on a C57BL6 and mixed athymic nude background were previously described[30,34]. To generate TEM8 KO (or floxed) mice on a BALB/c or FVB mice, C57BL6/TEM8 KO (or floxed) mice were backcrossed at least 10 generations to BALB/cAnNCrl (Charles Rivers) or FVB/NJ (The Jackson Laboratory). The following TEM8 strains developed in our laboratory are available from the Jackson Laboratory: B6-TEM8-flox (B6N.Cg-Antxr1tm1.1Bstc/J, Stock No. 037486), FVB-TEM8-flox (FVB.Cg-Antxr1tm1.1Bstc/J, Stock No. 037488), or BALB/c-TEM8-flox (C.Cg- Antxr1tm1.1Bstc/J, Stock No. 037490). The Tie2-cre strain (B6.Cg-Tg(Tek-cre)12Flv/J, stock number: 004128), Fsp-cre strain (B6.Fsp1.cre BALB/c-Tg(S100a4-cre)1Egn/YunkJ, stock number: 012641), mTmG reporter (B6.129(Cg)-Gt(ROSA)26Sortm4(ACTB-tdTomato,- EGFP)Luo/J, stock number: 007676), TEM8 mutant strain with a deletion of the transmembrane domain (Lepp-Del-TM, Antxr1tm1.2Lepp/J, stock number: 027705) and MMTV-PyMT strain (FVB/N-Tg(MMTV-PyVT)634Mul/J, stock number: 002374) were obtained from The Jackson Laboratory. Tamoxifen inducible VE-Cadherin(PAC)-CreERT2 (VE-cad-cre strain) was a kind give from Dr. Ralf Adams[42]. To perform human tumor xenograft studies using conditional TEM8 KO

mice, strains with TEM8 null or floxed alleles or cre transgenes were crossed to a SCID strain [B6.CB17-Prkdcscid/SzJ mice; The Jackson Laboratory (Stock Number: 001913)].

### TEM8-E150V knock-in gene targeting

Tem8-E150V Knock-In (KI) mice were generated by targeting the murine TEM8 (Antxr1) locus with CRISPR/Cas9. Two specific sgRNAs were designed to target the E150 codon of TEM8 (sgTem8 #1: 5′-TCATCATCGCGTTGACGGAT-3′, PAM = GGG; sgTem8 #2: 5′-GACGGATGGGGAGCTGCACG-3′, PAM = AGG). A single-stranded DNA donor (Integrated DNA Technologies) containing the E150V mutation (GAG codon to GTA codon) and disrupting the PAM sequence sgRNAs allowed homology-directed integration. We also generated a new ScaI restriction site (AGTACT) which was used during the early screening phase to identify targeted mice. TEM8-E150V ssDNA oligo sequence (180 nt) [lowercase indicates intronic region; uppercase indicates exonic region; bold indicates sequence modifications; ScaI restriction site is highlighted in italic]: tattcctgtcggttcaggatgactgacagggtcttg ctgtttcgagcagGATACAGGACGGCGAGCGTCATCATCGCGTTGACGGA TGG*AGTACT*GCACGAA**A**GACCTCTTCTTCTACTCAGAGAGGGAGgtgagt ggcagccactgggtctcaggaggaaacgggacctagccctcag.

sgRNAs were generated in vitro using MEGAshortscript T7 transcription kit (Thermo Fisher Scientific; AM1354). gRNAs were then purified using a MEGAclear kit (Thermo Fisher Scientific; AM1908). Cas9 mRNA, 100 ng/μl (TriLink Biotechnologies; L-7606), sgTem8 #1 or sgTem8 #2 (50 ng/μl) and the TEM8-E150V ssDNA donor (50 ng/μl) were microinjected (pronuclear microinjection) at the one-cell stage into mouse zygotes obtained from C57BL/6Ncr × B6D2F1/J mice to generate Tem8-E150V KI animals. B6-Tem8-E150V KI mice are available from The Jackson Laboratory [B6N.Cg-Antxr1em1Bstc/J, Stock No. 037492].

### Genotyping

E150V mice were genotyped by performing two separate PCRs, one for the TEM8 WT allele and one for the E150V knock-in allele. The WT allele was detected using the PCR primers WT-F: 5′- CCTGACCATCACTG-GAACTG-3′, WT-R: 5′-GAGGTCCTCGTGCAGCTCC-3′. The mutant allele was detected using the PCR primers E150V-F: 5′-CGCGTTGACGGATG-GAGTA-3′ and E150V-R:CAACGACTATCGGTTTTGTCTG. For PCR amplification of the IL2 positive control, the primers IL2-F: 5′-CAGAG-GACAGGGAGTGGTAAAAGC-3′ and IL2-R: 5′-GTAGGTGGAAATTCTAG-CATCATCC-3′ were included in each reaction. Expected amplicon sizes are 305 bp, 192 bp, and 444 bp for the wildtype, E150V knock-in, and IL2 genes respectively.

### Tumor growth

WT or TEM8-KO littermates derived from Tem8 heterozygous intercrosses were randomly assigned to experimental groups. Both males and females were used for tumor studies and the age of the mice at treatment start varied from 3 to 5 months unless stated otherwise. The strain of mice used for each experiment (C57BL6, FVB, BALB/C, athymic nu/nu, SCID) is indicated in the figure legends. Because C57BL6 TEM8 KO mice sometimes develop misaligned incisors as they age[30], to prevent malnutrition offspring from heterozygous intercrosses were provided wet feed, and teeth of offspring clipped starting at 2 months of age. For studies with VE-cad-cre lines, mice between 4–6 weeks of age were treated with tamoxifen (40 mg/kg) dissolved in sunflower oil daily for five consecutive days per week for two weeks. E0771 and 4T1 breast cancer cells were injected orthotopically into the mammary fat pad and other solid tumors were implanted subcutaneously. For spontaneous tumor growth, fully inbred FVB/TEM8+/- mice were crossed with transgene (Tg+) positive FVB/MMTV-PyMT mice, and FVB/TEM8+/-; Tg+ offspring were crossed with FVB/TEM8+/- mice to obtain FVB/TEM8+/+; Tg+ and FVB/TEM8−/−; Tg+ littermates for the tumor studies. Tumors were measured at each mammary gland, and total tumor

volumes recorded for each mouse. Subcutaneous and mammary fat pad tumors were measured by caliper, and tumor volumes were calculated using the formula $L \times W^2 \times 0.5$ and presented as mean ± s.e.m. Orthotopic liver metastases were measured using BLI. BLI of living animals was captured with an IVIS Spectrum In Vivo Imaging system (PerkinElmer) using Living Image advanced in vivo imaging software (4.3.1.0.15880) (PerkinElmer). For therapeutic studies, mice were sorted into groups containing an equal average tumor size (usually ~50–80 mm³) immediately prior to antibody administration. Mice were excluded from sorting if their tumors were less than half the average tumor size for the group or more than double the average tumor size. Tumor measurements were taken by technicians blinded to the objectives of the study. The survival endpoint was reached when mice became moribund or lost more than 20% of their body weight or individual tumors reached the maximal size of 4000 mm³ permitted by our institutional ACUC. In some cases, this limit has been exceeded the last day of measurement and the mice were immediately euthanized. Mice were treated intraperitoneally with vehicle (PBS), TEM8 antibodies, or free drug at the doses and schedules indicated in the figure legends.

### Intrasplenic production of liver metastasis
To produce liver metastasis by intrasplenic injection, the spleen was exteriorized through a left lateral incision. The tumor cell suspension was injected over a period of 5 min, allowing entry into the portal circulation, after which the spleen was removed.

### Cancer cell ATP analysis
10,000 PC9 and SW620 cancer cells were seeded in wells of a 96-well plate in 100 µl of DMEM media with or without 4 mM glutamine containing high (10%) or low (0.5%) dialyzed FBS and cultured at 37 °C for 48 h. After 48 h, 50 µl of detergent and 50 µl of substrate solution were added to the cells for ATP analysis following the manufacture's recommended protocol (AbCam, ab113849). ATP levels were measured using luminescence in a ClarioStar (BMG Labtech) multipurpose plate reader.

### Purification of TEM8-AP fusion proteins
Conditioned media from 293 stable transfectants expressing the extracellular domain (ED) of mouse or human TEM8 fused to alkaline phosphatase were collected, filtered, adjusted to pH 7.7 with NaH₂PO₄ and supplemented with 1% Triton-X 100, 150 mM NaCl and 3 mM Imidazole. His-tagged AP-fusion proteins were captured by Ni-NTA agarose beads (1:200 v/v; Qiagen), purified according to the manufacture's protocol, and dialyzed into PBS. SDS-PAGE analysis followed by Coomassie blue staining revealed that AP-fusion proteins were at least 95% pure.

### TEM8-AP ELISA
Rat tail Collagen I was isolated from frozen rat tails[66] in acetic acid (Sigma Aldrich) and stored as crude extract. Fresh reconstitution of rat tail collagen I in 0.1 M acetic acid was performed prior to each ELISA experiment. Stock solutions of 1 mg/mL rat tail collagen were diluted to a 2× final concentration in 0.1% acetic acid, then solid phased onto UltraCruz High Binding ELISA Multiwell Microplates (Santa Cruz Biotechnology) by diluting to desired concentration (i.e., 200 µg/mL) in 50 mM Tris-HCl (pH 8). pH neutralization allows fibrillar matrix formation and attachment to wells[38]. Plates were incubated at 37 °C for 2 h and washed four times in D-PBS containing calcium and magnesium and 0.5% (w/v) bovine serum albumen (PBS + CMB). TEM8-AP fusion proteins were purified as described above and added at the indicated concentrations in PBS + CMB to col1 films and incubated at 4 °C overnight with shaking. After overnight incubation, plates were washed four times in PBS + CMB and AP activity was detected using 1-Step PNPP Substrate Solution (ThermoFisher Scientific, cat# 37621).

Absorbance at 405 nm was measured with a ClarioStar (BMG Labtech) multipurpose plate reader. In later assays, to increase sensitivity, the PBS + CMB buffer was substituted with DMEM-F12/0.5%BSA/1.0% Tween 20 solution and the AP proteins were detected using HRP-anti-PLAP (Santa Cruz Cat# sc-47691) followed by absorbance reading at 450 nm.

### CHO ECM binding ELISA
1 µg of collagen I (Sigma, C3867), collagen IV (Sigma, CC076), collagen VI (Corning, 354261), laminin (Sigma, L2020), protective antigen (List Biological Labs,174 A) and BSA in 100 µl PBS were coated for 2 h at room temperature on 96-well tissue culture plates. Excess fluid was then removed from the coated surface and plates were allowed to air dry overnight. After rinsing plates with CHO media (Hams F12 containing 10% FBS) cells were added in triplicates at 25,000 cells/well in media and allowed to adhere for 1 hour at 37 °C, after which cells were gently washed twice with CHO media to remove unbound cells. Alamar blue was added to measure relative cells bound using a ClarioStar multipurpose plate reader.

### Soluble FITC assay
To measure cellular processing of collagen hydrogels, cells were embedded within collagen hydrogels prepared from FITC-Col labeled at a 1:1 molar ratio (Chondrex). In brief, cells were detached, washed twice in PBS + CM, and resuspended at $1 \times 10^6$ cells/mL in culture media. Collagen hydrogels were formed by neutralizing FITC-Col in 10× culture media supplemented with sterile 1 N NaOH (Sigma Aldrich) and 7.5% NAHCO₃ (Sigma Aldrich) in a rapid manner, adding cells, then plating in 96-well plates. Empty wells consisting of collagen hydrogels with no cells served as a positive control. After 45 min, 100 µL of media was added on top of collagen hydrogels which were incubated overnight at 37 °C with 5% CO₂. The next day, 50 µL of culture supernatant was removed to a black-walled, clear bottom 96-well plates (ViewPlate, PerkinElmer, Cat# 6005182) and FITC fluorescence from degraded collagen I-FITC hydrogels was measured with a ClarioStar multipurpose plate reader.

### FITC collagen internalization assay
Fluorescently tagged collagen uptake was quantified using a modification of the previously described protocol[67]. Briefly, cells were plated overnight on a 24-well plates. The next day a 50 µg/ml stock of FITC-Col was prepared by mixing 500 µL of collagen type 1-FITC (1 mg/mL Chondrex cat# 4001) with 9.5 mL media, incubating 20 min on ice, then adding 10 µL of 1 N NaOH to neutralize. CHO cell medium was removed from 24-well wells, replaced with new stock medium containing FITC-Col, and cells were cultured at 37 °C overnight. To block collagen internalization, in some experiments TEM8 antibodies were added (50 µg/mL final concentration) to both culture media at the time of cell plating, and to the FITC-Col stock solution. The next day, media on top of the collagen gel was removed, cells were trypsinized, washed in PBS, and 0.02% trypan blue was added to quench extracellular FITC immediately prior to performing flow cytometry.

### FITC-Col gel degradation assay
To form a gel, acid-solubilized FITC-Col (Sigma, cat# C4361) stock solution was diluted on ice to a 0.5 mg/mL gel solution by mixing with 0.1 volume of ice-cold 10× DMEM and neutralized using 1 N NaOH. After polymerizing on a 35 mm diameter glass-bottom plate (Mattek, cat # P35GC-1.5-14 C) for 45 min at 37 °C, the gel was rinsed with PBS, and cells were plated in complete media and incubated at 37 °C for 24 to 48 h. In some experiments, to block lysosomal uptake 50 mM NH₄Cl (Sigma, Cat# AX1270) was added for 24 h prior to visualization. On the day of imaging, Hoechst 33342 (ThermoFisher, cat# 62249) was added for 30 min to stain nuclei, and CellMask deep red plasma membrane stain (ThermoFisher, cat# C10046) for 5 min to stain cell surface

membranes. Images were captured with a Zeiss LSM 780 confocal microscope and analyzed using Zen 2.6 software.

## TEM8 immunohistochemistry

FFPE sections of human colon cancer were deparaffinized, treated with proteinase K, Dual Endogenous Enzyme-Blocking Reagent, biotin block (Dako), and then blocked with 1% blocking reagent (Roche) in TBS (100 mM Tris [pH 7.5], 150 mM NaCl) plus 1% Triton-X 100. Sections were stained with rabbit anti-human TEM8 (c37; Epitomics) for 2 h at room temperature, followed by signal amplification (Vectastain ABC HRP Kit; Vector Laboratories).

## Cell viability assays

For nutrient deprivation studies, TSCs or luciferase-tagged cancer cells were cultured overnight in complete DMEM media (ThermoFisher Cat# 11960) on 96-well microplates (TSCs on Corning 96-well plate, Cat# 3596 and Cancer cells on Corning luminescent white bottom plate, cat# 354651). The next day, media was removed and replenished with DMEM containing high (10%) or low (0.5%) dialyzed FBS (HyClone, SH30079.03; 10,000 MW cutoff), and supplemented with (4 mM) or without glutamine, and with (0.5 mM) or without proline as indicated in the figures. In some experiments, as indicated in the figures, 40 µg/ml col1 (Millipore, 08-115 in 0.02 N acetic acid, pH 3.6) was added directly to wells and neutralized by adding 0.02× volume of 1 N NaOH. To measure cancer cell viability 48 h later, media was removed and replaced with luciferin in 1XPBS buffer (Goldbio, #LUCK- 1 G) and luciferase measured using a ClarioStar (BMG Labtech) multipurpose plate reader. Because TSCs were not labeled with luciferase, their viability was measured 48 h post-treatment using an alamarBlue assay (ThermoFisher, cat# DAL1025) and the ClarioStar plate reader.

## Coculture assays

Before coculture, TSCs were maintained at 32 °C. Upon coculture, TSCs and cancer cells were trypsinized, counted, mixed in a 1:5 (cancer cell:TSC) ratio, and plated in 50:50 DMEM/RPMI supplemented with 10% FBS on a 96-well microplate at 37 °C overnight. The next day, medium was replaced with modified nutrient-deprived medium with or without 40 µg/ml collagen I as indicated. For antibody blockade, 50 µg/ml of m830 antibody was preincubated with CHO or CHO-TEM8 cells on ice for 30 min prior to mixing at a 1:5 ratio with cancer cells overnight and replenished the next day along with the other media modifications. For L-asparaginase treatment, 2 mU/mL L-asparaginase (Sigma, A3809) was added to the modified nutrient-deprived medium with or without 40 µg/ml collagen I as indicated. After 48 h at 37 °C, the viability of luciferase-tagged cancer cells was measured using luciferin as described above.

## Lentiviral particle production and TSCs target gene knockdown

Lenti-X 293 cells (Takara, 632180) were transfected with pMD2G (Addgene, 12259), psPAX2 (Addgene, 12260), and shRNA carrying plasmids (Sigma SHC202 for non-specific knockdown; TRCN0000075984 and TRCN0000309816 for GLUL knockdown; TRCN0000031905 and TRCN0000295140 for PEPD knockdown; TRCN0000256401 and TRCN0000256403 for PRODH knockdown) via 1:2:2.5 ratio using Lipofectamine 2000 (Invitrogen, 11668019) according to the manufacturer's manual. After 24 h transfection, medium was replenished with fresh DMEM medium containing 10% FBS. Lentivirus-containing media was filtered (0.45 µm), mixed 1:1 with fresh media containing 10 µg/ml polybrene, and used to infect TSCs maintained at 32 °C. To increase knockdown efficiency infections were repeated at least twice per day. TSCs were recovered in DMEM containing 10% FBS for one day and then split into media containing 2.5 ug/ml Puromycin and cultured at 37 °C for 48 h. Cells were trypsinized and plated in co-culture experiments, or collected for molecular analysis as indicated.

## Western blotting

Cancer cells or TSCs were treated with different nutrient-deprived conditions with or without 40ug/ml collagen I (Millipore, 08-115) and antibodies against TEM8 as indicated. After treatment, cells were washed with 1xPBS (Corning, 21-031-CV) and detached using a cell scraper. Cells were pelleted and resuspended in lysis buffer (50 mM Tris-pH7.5, 1% Triton-X and 150 mM NaCl). Following clarification, soluble lysates were mixed with an equal volume 2× Laemmli sample buffer (Bio-Rad, 1610737) containing 1% beta- mercaptoethanol (Sigma:M6250) and boiled at 95 °C for 10 min, run on a 4–15% TGX SDS-PAGE gel (Bio-Rad, 5671084) and transferred to PVDF membrane (Millipore: IPVH00010).

The antibodies used for Western Blotting were PEPD (Santa Cruz: sc-390042), GLUL (Cell signaling: #80636), PRODH (Santa Cruz: sc-376401), TEM8 (C37 in house), CMG2 (1H8, a gift of Stephen H. Leppla), and β-actin (Santa Cruz: sc-47778). After labeling with primary antibodies followed by HRP-conjugated secondary antibodies, target proteins were detected by ECL Chemiluminescence Imaging (Pierce Cat#: 34076) according to the manufactures protocol. To verify TEM8 absence in TSC-KO cells (Supplementary Fig. 7c), a long exposure was used, while a short exposure was needed to demonstrate the elevation in TEM8 upon nutrient stress (Fig. 5).

## Immunofluorescence (IF) staining

For co-IF staining of tumors from mTmG reporter mice, frozen cryosections were fixed with methanol:acetone (50:50) and stained with a chicken anti- GFP (AbCam, ab13970), rabbit anti-desmin (AbCam, ab15200) and rat anti-CD31 (Santa Cruz SC18916). The GFP signal was amplified using FITC goal anti-chicken (Jackson Immunoresearch) followed by Alexa 488 goat anti-FITC (Thermo A-11096), the desmin signal was amplified using biotinylated donkey anti-rabbit (Jackson ImmunoResearch 711-065-152) followed by Alexa 350 Streptavidin (Thermo Fisher, S11249), and the CD31 signal was amplified using Alexa 647 donkey anti-rat (Jackson ImmunoResearch, 112-605-175). For co- IF staining of vessels and CAFs in tumors, frozen cryosections were fixed with 1% paraformaldehyde (PFA)/PBS and stained with mouse anti-a-Smooth Muscle Actin (SMA) antibody (Sigma cat# A5228) or m825 human anti-TEM8.SMA signal was detected using biotin-donkey anti-mouse IgG (Jackson ImmunoResearch, cat# 715-065-151) secondary antibody followed by Texas red-streptavidin (Vector Laboratories, cat # SA-5006-1) while TEM8 signal was detected using Fluorescein (FITC) goat anti-human IgG (Jackson ImmunoResearch cat# 109-095-088) followed by Alexa-488 goat-anti-FITC (ThermoFisher cat# A-11096). During the last staining step, an Alexa-647 anti-mouse CD31 antibody (Biolegend cat# 102516) was added, and nuclei were counterstained with DAPI (Hoechst 33258, ThermoFisher cat# H3569) prior to mounting. For Ki67 staining, frozen cryosections of PyMT tumors were briefly fixed with 1% PFA/PBS and stained with a rabbit anti-Ki67 antibody (Abcam, ab15580) followed by biotin-labeled donkey anti-rabbit (Jackson ImmunoResearch 711-065-152) and streptavidin Texas red (Vector Labs SA-5006). During the last staining step nuclei were counterstained with DAPI and the percent Ki67 positive nuclei were calculated using Image J software. For co-IF staining of col1 and TEM8 in tumors, frozen cryosections were briefly fixed with 1% PFA/PBS. Human colon cancer liver metastasis was stained with a mouse anti-collagen (Southern Biotech, 1441-01) followed Alexa 594 labeled goat anti-mouse (Jackson ImmunoResearch, 715-585-151) while TEM8 was stained with rabbit anti-TEM8 (c37, AbCam ab241067) followed by FITC-labeled anti-rabbit (Jackson ImmunoResearch, 111-095- 144) and 488-linked goat anti-FITC (ThermoFisher, A11055). MC38 and HCT116 liver metastasis were stained with a rabbit anti-collagen (Rockland 600-401-103) followed by Alexa 594 labeled donkey anti-rabbit (Jackson ImmunoResearch, 711-585-152) while TEM8 was stained with human anti-TEM8 (m825, in house) followed by FITC-labeled anti-human (Jackson ImmunoResearch, 109-095-088) and 488-linked

donkey anti-goat (ThermoFisher, A11055). During the last staining step nuclei were counterstained with DAPI. For collagen staining of PyMT tumors, frozen cryosections were fixed with 2% PFA/PBS, then stained overnight at 4 °C using a collagen binding probe, EGFP-CNA35, that was produced and purified from bacteria using the expression vector pET28a-EGFP-CNA35 (Addgene #61603)[38]. CNA35 has been shown to bind collagens I, III, and IV[68]. Images were captured with a Zeiss LSM 780 confocal microscope and analyzed using Zen 2.6 software.

### Live cell staining

Before seeding TSCs, glass bottom culture dishes (MatTek, P35GC-1.5-14-C) were coated with a collagen gel containing NaOH neutralized 50 µg/ml FITC-labeled bovine type I Collagen (Chondrex: 4001) in DMEM (Corning 10-013-CV). Immediately after gel formation, 50,000 TSCs were seeded in DMEM on top of the collagen gel and incubated overnight at 37 °C. The next day, lysotracker Red-DND-99 (Invitrogen: L7528), CellMask™ Deep Red (Invitrogen: C10046) and NucBlue (Hoechst 33342, Invitrogen: R37605) were added to culture medium at the manufactures recommended concentration for 15 min at 37 °C. For TEM8 staining, 20 mM $NH_4Cl$ was added to the TSC culture medium when cells were plated on the FITC-Col gel-coated glass bottom culture dishes overnight. The next day, cells were stained on ice for 1 h with biotinylated m830 antibody (biotinylated using the EZ-Link™ Sulfo-NHS-Biotin kit, ThermoScientific: 21326), followed by Alexa647 conjugated-streptavidin (Invitrogen: S32357) for 30 min. After staining, cells were incubated at 37 °C for ~30 min to enable antibody internalization, and nuclei were stained with NucBlue. Images were captured using a Carl Zeiss LSM780 microscope.

### TEM8 antibody production and purification

The m830 antibody was derived from a yeast display library constructed using a collection of human antibody gene repertoires[69] and those from more than 50 additional individuals. Due to the in vitro stochastic pairing of VH and VL repertoires, this library is not subject to tolerance mechanisms found in normal immune responses and allowed the generation of antibodies against regions of the TEM8 ECD that are 100% conserved between mouse and human. In vitro selection of the yeast display library involved multiple rounds of sequential panning on biotinylated, purified recombinant human or murine TEM8(ED)-AP and TEM8-Fc fusion proteins. Biotinylated recombinant TEM8 proteins were incubated with $5 \times 10^{10}$ cells from the antibody library in PBS-BSA (PBS containing 0.1% BSA) for 2 h, washed with PBS-BSA, and captured with streptavidin-conjugated microbeads from Miltenyi Biotec using the AutoMACS System. The sorted cells were amplified, and the panning was repeated. After further scFv validation, m830, one of the lead antibodies from this screen, was converted into a full-size human IgG1. m830 was collected from culture supernatants grown in serum free medium and purified by protein A chromatography. Antibody preparations for in vivo studies possessed less than 5% aggregates and had endotoxin levels below 1 EU/mg.

### Flow cytometry

TEM8 positive or negative cells were trypsinized, rinsed in cold PBS-BSA, and labeled with m830 human IgG in PBS-BSA at 4 °C. Next, cells were rinsed, incubated with FITC-conjugated goat anti-human IgG antibodies, and then rinsed again. Analysis was performed on a FACSCalibur Flow Cytometer or a LSRII (BD) and data processed using FlowJo (v10).

### ELISA to detect serum TEM8

To detect soluble TEM8 (sTEM8) in human serum, m825 anti-TEM8 antibody[21] was solid phased onto 96-well plates, blocked with superblock, and serum samples, or TEM8-AP positive control protein was added. The sTEM8 was detected with SB5 anti-TEM8[27], followed by biotin-labeled goat anti-mouse Fcγ (Jackson ImmunoResearch, 115-

065-164), and streptavidin HRP (Jackson ImmunoResearch, 16-030-084). HRP activity was quantified using Ultra TMB-ELISA Substrate Solution (Thermo Scientific, 34028) which was detected at 450 nm on a ClarioStar plate reader.

### ELISA to measure m830 in mouse serum

To detect soluble m830 in mouse serum from TEM8 WT and KO mice, AP-TEM8 was solid phased onto 96-well microplates (Beckman Coulter Cat# 609844). After rinsing wells with wash buffer (PBS/Tween 20 (0.05%)), wells were blocked with SuperBlock (Thermo Scientific Cat# 37515) and then pre-diluted samples and standards added to wells and incubated at room temperature. Bound human m830 was detected using biotin goat anti-human antibodies (Jackson 109-065-088) followed by Streptavidin HRP (Jackson 016-030-084). After rinsing again, Ultra TMB-ELISA Substrate Solution (Thermo Scientific, 34028) was added and detected at 450 nm on a ClarioStar plate reader.

### Antibody affinity measurements

m830 Fab was generated from full-size IgG1 using the Pierce Fab Preparation Kit (Thermo Fisher Scientific). Surface plasmon resonance was used to measure binding affinity of the Fab to the TEM8 ECD on a BIAcore X100 instrument (GE Healthcare). Purified mAP-TEM8 and hAP-TEM8 fusion proteins were diluted in 10 mM sodium acetate buffer (pH 5.0) and immobilized on a CM5 biosensor chip using an amine coupling kit. The running buffer was HBS-EP (10 mM HEPES, pH 7.4, 150 mM NaCl, 3 mM EDTA, 0.05% surfactant P20). The Fabs diluted with the running buffer were allowed to flow through the cells at concentrations ranging from 0.05 nM to 500 nM. After 10 min of dissociation, the chip was regenerated with 10 mM acetate buffer, pH 4.0. The data were fitted with a 1:1 binding model, and the dissociation rate constant was estimated with BIAevaluation software (Biacore).

### Amino acids analysis by LC/MS/MS

$3 \times 10^6$ WT-TSCs and KO-TSC were cultured in 60 mm tissue culture dishes in DMEM medium supplemented with 10% FBS (Avantor Seradigm: 97068-085) at 32 °C overnight. The next day, TSCs were rinsed with glutamine-free DMEM (DMEM-GF, Corning: 25-005-CI), and 2.5 ml of 1 mg/ml collagen I (Millipore) was mixed in DMEM-GF pre-mixed with 0.5 mL 1 N NaOH (Fisher: SS2551) (Millipore: 08-115) on ice and layered on top of cells for 10–20 min at 37 °C to create a collagen I gel. Afterwards, 2.5 ml DMEM-GF media supplemented with 1% dialyzed FBS (Hyclone: SH30079.03) was added on top of the collagen I gel. For m830 anti-TEM8 antibody treatment, WT-TSCs were mixed at the start of the assay with 100 µg/ml m830 in DMEM prior to plating on 60 mm tissue culture dishes overnight. After rinsing with DMEM-GF, TSCs were overlayed with a NaOH-neutralized collagen I/DMEM-GF mixture containing 100 µg/ml of m830. 2.5 ml DMEM-GF supplemented with 1% dialyzed FBS and 100 µg/ml m830 was then placed on top of the collagen I gel. Cells were incubated at 37 °C for 48 h, after which the conditioned media were collected and centrifuged at maximum speed using a benchtop centrifuge. Clarified supernatant was collected, frozen, and sent for analysis to the NCIs Center for Cancer Research (CCR) Protein Characterization Laboratory (PCL) Mass Spectrometry Center. For LC/MS/MS, the amino acid and the isotopic internal standard (IS) amino acid mixtures were obtained from Cambridge Isotope Laboratories and all other chemicals were from Sigma. The amino acid and the IS stock solutions were prepared in water. The calibration standard solutions were prepared by diluting the stock solution with water. The IS working solution (20 µM) was prepared in sodium carbonate (80 mM) and the benzoyl chloride solution (1%, v/v) in acetonitrile. Amino acids in the culture media were determined by LC/MS/MS after chemical derivatization with benzoyl chloride as previously described[70] with slight modification. In a 500 µL polypropylene tube, 20 µL of media (or standard solutions) were mixed with 20 µL of IS and 40 µL of benzoyl chloride for several minutes. 20 µL of the reaction

mixture was mixed with 20 μL of water and the diluted solutions were transferred to polypropylene sample injection vials. LC was performed with a Shimadzu 20AC-XR system. Separation was achieved at 50 °C with a 2.1 × 100 mm, 1.8 μm T3 column (Waters). Mobile phase A was 0.15% formic acid with 10 mM ammonium formate in water and mobile phase B was acetonitrile. The flow rate was 300 μL/min, and the injection volume was 3 μL. The amino acids were separated with a gradient from 5% to 65% B in 10 min. MS/MS was performed with a TSQ Quantiva triple quadrupole mass spectrometer (Thermo Scientific) operating in positive SRM mode. The amino acid concentrations in the media were determined by linear calibration curves with $1/x$ weighting generated by the Thermo Xcalibur Quan Browser software (version 4.3). The curves were constructed by plotting the peak area ratios vs. standard concentrations. The peak area ratios were calculated by dividing the peak areas of the amino acids by the peak areas of the corresponding IS.

## Quantification and statistical analysis

A Students $t$ test was used to calculate differences in tumor volumes between two groups. A Students $t$ test was used to calculate differences in tumor volumes between two groups. For comparison of three or more groups, a one-way analysis of variance was used with a Tukey's test. For Kaplan Meier survival analysis, a Log-rank (Mantel-Cox) test was used to compare each of the arms. All measurements were taken from distinct samples. Differences between two groups were presented as the mean ± s.e.m. or mean ± s.d. as noted in Figure Legends. Experimental sample numbers ($n$) are indicated in Figure Legends. All tests were two-sided and $p$ values < 0.05 were considered statistically significant. All statistical analysis was performed with GraphPad Prism 9.2.0.

## Reporting summary

Further information on research design is available in the Nature Portfolio Reporting Summary linked to this article.

# Data availability

The data that support the findings of this study are available within the Article, Supplementary Information, or Source Data file. Source data are provided with this paper. The mouse strains generated in this study have been deposited to the Jackson Laboratory under Stock No. 037486 (B6-TEM8-flox), Stock No. 037488 (FVB-TEM8-flox),Stock No. 037490 (BALB/c-TEM8-flox) or Stock No. 037492 (B6-Tem8-E150V KI). Some materials may require requests to collaborators and/or agreements with various entities. Materials that can be shared will be released via a Material Transfer Agreement. Source data are provided with this paper.

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

## Acknowledgements

We thank Dr. James M. Phang for expert advice and helpful discussions. This work was supported by the Nutritional Science Research Group, Division of Cancer Prevention, NCI/NIH, and by a Cooperative Research and Development Agreement (CRADA) between BioMed

Valley Discoveries and the NCI. Federal funds were also provided by the Center for Cancer Research (CCR), part of NCI's intramural research program, NIH, Department of Health and Human Services (DHHS). The content of this publication does not necessarily reflect the views or policies of the DHHS nor does mention of trade names, commercial products, or organizations imply endorsement by the US Government.

## Author contributions

Conceptualization, K.-S.H., J.M.D., C.S., N.J.E. and B.S.C.; Methodology, K.-S.H., J.M.D., C.S., F.T.-A., E.Z., J.R. and F.C.; Investigation, K.-S.H., J.M.D., C.S., L.Y., M.B.H., K.M., S. Seaman,Y.F., E.M.L., R.K., F.T.-A., E.Z., Z.Z., P.B., F.C. and B.S.C.; Writing—Original Draft, K.-S.H. and B.S.C.; Writing—Review & Editing, all authors; Funding Acquisition, S. Saha, N.J.E., and B.S.C.; Supervision, K.-S.H., J.M.D., Z.Z., X.M.Z., D.S.D., L.T., and B.S.C.

## Funding

## Competing interests

S. Saha, X.M.Z., E.Z., Z.Z., D.S.D., and B.S.C. are inventors of intellectual property related to TEM8 antibodies, and B.S.C. and D.S.D. have received research support through a CRADA with BVD. The remaining authors declare no competing interests.
