## [Peer Review File · Nature Communications]

Cancer cell survival depends on collagen uptake into tumor-associated stromaREVIEWER COMMENTS

Reviewer #1 (Remarks to the Author):

The paper focuses on TEM8 as a collagen-binding protein and discovers its role in tumor growth under starvation conditions. Expression of the receptor appears to be important for tumor growth. The receptor does in fact bind collagen I and IV in the ELISA assay, with a binding motif similar to that of integrin. In cells, the expression of TEM8 does not appear to be dependent on oxygen levels, but rather on the available nutrients - specifically glutamine. Finally, the authors develop an antibody against TEM8 and test it in mouse models. alphaTEM8 alone has a significant effect, which is further enhanced by combination with antibodies that have already been developed for the respective tumors (Fig. 7). This result shows that the findings can be in principle harnessed in new therapeutic strategies targeting TEM8.

The study comprises a larger set of experimental techniques that jointly support the conclusions very well.

I have several concerns regarding individual experiments such as controls that should be addressed.

- Which motif could be bound in collagen (see also Fig. 3)?
- For the fluorescence microscopy in Fig. 3, one could still consider carrying out a quantification.
- What would happen if you perform a knock-in with a tumor that has not shown TEM8 dependence (see Fig. 4f)? Maybe a useful negative control?
- In Fig. 5f there is no comparison to healthy tissue.
- Collagen staining for Fig. 6 d+j would be helpful to see if the collagen is actually degraded.
- Was serum glutamine measured in the TSC/SW620 co-culture to verify the mechanism?
- Would a radiolabeling experiment with glutamine make sense to show that the amino acid is actually converted into energy? Or a simple ATP assay (Fig. 6)?
- Also in Fig. 7: Effect of the antibody therapy on collagen in tumor tissue would be interesting.

Reviewer #2 (Remarks to the Author):

In this study, Hsu et al. focus their research on TEM8, they show, using several mouse tumor models, that this cell surface protein, strongly expressed in tumor associated stroma, is crucial for tumor growth. Based on their previous publication showing high expression of TEM8 in stromal cells such as cancer-associated-fibroblasts (CAFs), tumor associated endothelium and pericytes, they describe the essential role of TEM8 in tumor stromal cell (TSCs) for collagen1 uptake and demonstrate that TEM8 expressing TSCs support tumor growth and this pro-tumoral role of TSCs depends on their ability to uptake collagen. In a second part of the manuscript, authors highlight that nutrient stress such as FBS depletion, glutamine restriction, induce TEM8 expression in TSCs, HMECs cells. In these nutrient restricted conditions, addition of collagen1 partly rescues TSCs survival and this rescue is TEM8 dependent. Moreover, authors show that co-culture of TSCs expressing TEM8 with tumor cells is required to observe tumor cell survival upon low FBS expression and collagen addition. From this observation authors suggest that TEM8-dependent collagen uptake by TSCs can further metabolically support cancer cells, through collagen-derived glutamine supply. Finally, using TEM8 specific antibodies to block collagen uptake, authors demonstrate in vivo that TEM8 activity is essential for tumor growth and metastatic burden and that combination of such TEM8 targeting with routine chemotherapy improves the efficacy of several chemotherapeutic agents in several tumor models.

Globally this work presents an interesting role of TEM8 in the crosstalk between TSCs and cancer cells and moreover demonstrates that the phenotype observed is common to several cancer types. Also, in this context the determinant role of collagen1 in tumor growth is also highlighted. Finally, authors propose relevant tools to target TEM8 and present strong in vivo validation on mouse tumor models for the use of TEM8 blocking antibodies in clinic. It remains that the demonstration of the metabolic support of TSCs-processed collagen1 to tumor cells is weak and requires solid validation that is not presented in the present manuscript. As authors include a huge amount of

data in this manuscript, use several models of tumor mouse models and strongly demonstrate the specific role of TEM8 in tumor growth, I suggest that authors should tone down their conclusions about this metabolic support of col1 and consider the following suggestions/requirements to improve the robustness of their work.

Major Comments :

Point 1: Figure1, 2: Some data in these figures show a delay in tumor growth (fig1 a,d,g,h, fig2c,d,f,g,h,j) more than a long term tumor repression, and others show a clear impact on tumor inhibition (fig.1b,e, fig2 k). Hence, authors should give a more robust conclusion in regards with data of these 2 figures:

1/ highlight conclusion about the effect of targeting TEM8 either in fsp+ or VE cad+ cells, as it appears that targeting TEM8 in fsp+ cells is more efficient in term of inhibition of tumor growth than in VE cad+cells,

2/ precisely mention when data show tumor delay vs a decrease in tumor growth,

3/ add some data showing the proliferative index of these tumors (ie their relative aggressiveness by Ki67 staining, especially for panel showing tumor growth delay), 3/ their ECM and col1 content by col1 IHC, trichrome staining and finally H/E staining.

In fig2: IHC or IF of FSP+cells in tumors TEM8fl/+ or fl/- would indicate CAFs/TSCs content of these tumors following TEM8 modulation.

Point 2: figure 3: to validate the uptake of Col1 by CHO-TEM8 cells, authors should show quantification of Col1 internalization by FACS, using soluble col1FITC. Also, IF showing Col1 internalization in lysosomes using co-if with LAMP1 marker is required. Finally co-IF of TEM8 and Col1 is missing.

Point 3: Figure 4c-e: As the work is focused on the role of TSCs in collagen uptake and its dependency on TEM8, some of the data showing TEM8-dependent col1 binding, uptake using various TEM8 mutants in panels c-e, should be validated on TSCs.

Figure 4f: Again H/E staining of these tumors is required. Also, IF of wt vs mutant TEM8 would be indicative of its localization in either CAFs or VE cells (co-staining with specific markers of these cells)

Point 4: Figure 5: Origin of TSCs as to be better defined than just citation of previous work in JCI. Please indicate/show whether these TSCs are TEM8 low, high, mixed as in Fig 5c TEM8 is low at basal expression. Are the TSCsWT used in fig6b different from the ones used in fig 5c and d as levels of TEM8 is different at basal level between these panels? Co-IF of TEM8 and CAFs is required in panel fig5f.

Point 5: Fig6: In the context of assessing the role of col1 in tumor cell survival upon nutrient stress, authors could evaluate the rescue of col1 addition in condition of low glutamine using dialyzed 10% FBS, to avoid striking reduction of important amounts of nutrient using low FBS. Blots showing levels of TEM8 in SW620 should be shown in this manuscript. This would add precision to the sentence "although cancer cell TEM8 status did not impact these responses". In panel fig6e data showing SW620 viability when co-cultured with TSCs in low FBS without addition of col1 would indicate the rescue effect of TSCs only on cancer cells in condition of low FBS and this point would allow comparison with the same condition with addition of Col1, in order to evaluate 1/the rescue of TSCs alone on cancer cell survival at low FBS and 2/the effect of col1 on cancer cell survival in condition of co-culture.

Point 6: As prolidase specifically allows proline release from dipeptides of proline-glycine from collagen, and GLUL produces glutamine from glutamate, neither PEPD nor GLUL are enzymes in the collagen degradation pathway nor they are key enzymes to be targeted to assess if col1 is converted into glutamine (please re-phrase the associated sentence in the manuscript). Hence, fig 6h and I data only show that proline release and glutamine production in TSCs cells support SW620 survival on collagen plates at low FBS and that only glutamine rescues the KO of both enzymes. Based on this observation, and on data from fig 6j, as glutamine from TSCs might derived from various metabolites other than proline, authors definitively have to KO PRODH enzyme to inhibit glutamine production from proline degradation to conclude that gln release from TSCs collagen-derived proline supplies cancer cells with glutamine pool (use of PRODH KO TSCs

co-cultures +/-col1, +/- gln/pro). Another way to demonstrate this would be proline tracing in TSCs cultures to follow carbon flux from proline derived col1 to glutamine and supply this labelled gln to cancer cells to assess the fate of these proline-derived carbon into cancer cells.

Point 7: figure 7: the effect of TEM8 on metastasis is surprising as metastasis microenvironment contains few and variable amounts of CAFs and have abundant supply of oxygen and nutrients as compared to primary tumors. Hence this raises questions about the role of TEM 8 in metastases as a driver of collagen-derived metabolic support. Data showing TEM 8 staining in metastasis are missing.

Minor points :

Minor Point 1: Fig1a-c: precisions of athymic nude mice models have to be added in mat and met section. Numbers of mice in groups of these panels have to be indicated as for other panels where it is missing.

Minor Point 2: Panel fig 1g should be placed in sup data. Panel fig 2i shows only UACC tumor not SW620, RENCA as mentioned in the text: is there a panel missing in this figure or in sup data?

Minor Point 3: Stats: indicate in figure legends to which comparison p value are associated with.

Reviewer #3 (Remarks to the Author):

Kuo-Sheng Hsu et al used mouse models and in vitro experiment to investigate the role of TEM8 in tumor development and cancer cell survival. The findings are interesting, results are convincing and controls experiments well performed. The manuscript lacks overall methods description and reagents references.

1. The authors use a TEM8 KO model in this manuscript: do the TEM8 KO mice show any phenotype like head or dental malformation as in GAPO patients? Same for conditional KO mice (Tie2-cre, VE-cad-CreERT2 and Fsp1-Cre)?
2. General use of collagen in the description of the results should be replaced by the appropriate collagen depending of the experiment performed. For example, page 7 (Figure 1f) the authors claim "increased collagen levels in tumors derived from TEM8 KO versus TEM8 WT animals" Please specify the collagen type (Collagen I only shown in this figure).
3. In figure 1h,I, the authors compare the percentage of tumor growth inhibition between WT and KO animals at 4 and 11 months. However, the comparison does not seem to be done at the same moment (Different day after inoculation) Please clarify or compare similar dpi please.
4. Figure 2: On page 9 the authors claims than TEM8+ vascular cells in the SW620 and RENCA models play minor roles but no data are shown for this. Please clarify.
5. The authors describe the presence of laminin and collagen IV in matrigel, can they provide a reference, do they know the amount of Collagen 1 or Collagen VI in matrigel
6. figure 3 bc, the amount and origin of collagens are described only for Collagen I, are the other collagens used at 200ug/ml? could the authors give references for collagen they used.
7. figure 3h, could authors give the concentration of the gradient of COL1 and COL6 they are using. Molecular weights markers should be added.
8. The authors use TEM8+ and TEM8- (and sometimes CMG2+ cells) to describe CHO overexpressing or not TEM8 or CMG2. Please use the correct name CHO, CHO-TEM8 (for CHO overexpressing TEM8) or CHO-CMG2 (for CHO-overexpressing CMG2). Showing endogenous levels of CMG2 and TEM8 in CHO could be useful.
9. Figure 3e : the binding assay is not described in the material and methods. How is it performed?
10. figure 4-a : D50, S54, T118, Q137 should also be included in the alignment among species, with E152 and H154.
11. Figure 4d, degradation of collagen is quantified in the culture supernatant but materiel and methods only describe IF staining. Please explain methods.

12. figure 5abcde: Molecular weights markers should be added.
13. Supplementary Figure 2: no description of arrow in the legend.
14. In some figure legends, the number of animals/replicates used is missing.

Reviewer #4 (Remarks to the Author):

In this study by Usu et al, the authors describe a novel mechanism regulating tumor growth through the uptake of glutamine released by fibroblasts upon collagen uptake and processing into glutamine. This is a very interesting study with substantial amount of data supporting the main hypothesis and multiple mouse models to verify that TEM8 in the tumor stroma regulates tumor growth through collagen consumption. The data presented open a new avenue of investigation into how regulation of the ECM by the stroma control tumor growth.

There are some aspects that could improve the study. Below I highlight my comments:

1) In figure 1, the authors show that in aged mice the effect of TEM8 KO in the stroma is larger than in younger mice. Have the authors tested if TEM8 decreases also with age?

2) Figure 2 a,b is a little confusing. It's my understanding that these images are control of the Tie2-Cre, VE-cad-cre and Fdp1-cre mice but the way it's written in the text gives the understanding that these images are from mice with TEM8 deleted in those compartments. The authors should show images of TEM8 staining in the mouse models upon TEM8 KO in the different compartments to verify depletion. Also, the images shown do not have scale bar.

In this figure the tumor growth panels are organized in a way that makes the figure difficult to understand. I will suggest to group the graphs in blocks, i.e all the FSp-cre (e,h,j,k) in the different cell lines under the same letter; for example (c). I think this will help understand the figure.

In figure 2i the authors mention in the text that SW620, RENCA and UACC models are shown but only UACC graph is shown in the figure.

3) In figure 3 the authors show that TEM8 binds to COL I and COL VI. Does TEM8 bind to other fibrillar collagens (II,III) or it's restricted to collagen I?

4) The authors use extensible CHO cells as a model of low TEM8 but later in the manuscript the authors show data with fibroblasts either WT or KO for TEM8. This seems to be a more representative model for the study the authors propose. Using these fibroblasts lines in the experiments described in Figure 3 and 4 will strength the paper.

It will also good to show a western blot of TEM8 in CHO and CHO-TEM8 cells to assess the level of overexpression.

5) In figure 3 and supp3 several experiments showing degradation of FITC-Col I are presented however no quantification is shown.

6) The metastasis experiments are interesting. However, performing those experiment in a spontaneous metastasis model will strength the claims. In the paper the authors use intrasplenic injection to deliver colon cancer cells to the liver. But, how is the metastatic cascade affected in the presence of m830? Does TEM8 blockage prevent dissemination and seeding of tumor cells in the liver? Does it prevent tumor growth at the metastatic sites? Experiments evaluating the metastasis potential of m830 in spontaneous metastasis model will strength this part of the paper.

Point-by-point response to the reviewers' comments

REVIEWER COMMENTS

Reviewer #1 (Remarks to the Author):

The paper focuses on TEM8 as a collagen-binding protein and discovers its role in tumor growth under starvation conditions. Expression of the receptor appears to be important for tumor growth. The receptor does in fact bind collagen I and IV in the ELISA assay, with a binding motif similar to that of integrin. In cells, the expression of TEM8 does not appear to be dependent on oxygen levels, but rather on the available nutrients - specifically glutamine. Finally, the authors develop an antibody against TEM8 and test it in mouse models. alphaTEM8 alone has a significant effect, which is further enhanced by combination with antibodies that have already been developed for the respective tumors (Fig. 7). This result shows that the findings can be in principle harnessed in new therapeutic strategies targeting TEM8. The study comprises a larger set of experimental techniques that jointly support the conclusions very well. I have several concerns regarding individual experiments such as controls that should be addressed.

Reviewer: - Which motif could be bound in collagen (see also Fig. 3)?

Response: We assume the reviewer is referring to the motif in TEM8 that is required for binding collagen, as collagen itself is a highly repetitive sequence primarily made up of repeating units of proline, hydroxyproline and glycine. In the TEM8 extracellular domain there is a Metal Ion Dependent Adhesion Site (MIDAS) motif in the center of the I-domain that is required for collagen binding. The MIDAS motif is shown in Figure 3a. By creating mutations in the MIDAS domain that block metal ion binding (D50A, S54A, T118A and D150A) we show that collagen binds to the MIDAS motif of TEM8. The corresponding residues in the MIDAS of integrins have also been shown to be critical for integrin binding to collagen. Based on homology modeling with the ITGA2-collagen crystal structure, we have also tested and identified two additional mutations near the TEM8 MIDAS motif that are important for collagen binding – E152 and H154. To address this concern, we have updated the text to clarify that collagen binding to TEM8 occurs at the MIDAS domain (page 5 and 10 of the revised manuscript). We have also added the other areas of the MIDAS domain to figure 4a, highlighting the residues that are conserved and required for TEM8 binding to collagen.

Reviewer: - For the fluorescence microscopy in Fig. 3, one could still consider carrying out a quantification.

Response: We agree with the reviewer that it is important to quantify the uptake of FITC-collagen into TEM8 positive cells. For this purpose, we developed a FITC Collagen Internalization Assay that allowed us to accurately measure the amount of collagen captured in cells by flow cytometry (see figures 4e and 7b and Extended Data figures 7b and 7d of revised manuscript). We also developed a FITC-Col Gel Degradation Assay which allowed us to measure the amount of FITC released into the supernatant of cells after collagen degradation (figure 4d). While the images in figure 3 help pinpoint the sites of collagen degradation, we feel our flow cytometry and ELISA assays provide the most accurate quantitation of collagen uptake and degradation.

Reviewer: - What would happen if you perform a knock-in with a tumor that has not shown TEM8 dependence (see Fig. 4f)? Maybe a useful negative control?

Response: While a TEM8 independent tumor could theoretically provide a useful negative control, so far, all tumors we have tested depend on host TEM8 for growth to some extent, and the level of dependence appears to increase as mice age (see for example, figure 1g and 1h of revised manuscript). Therefore, we do not have a suitable negative control tumor cell line at this time.

Reviewer: - In Fig. 5f there is no comparison to healthy tissue.

Response: We thank the reviewer for this suggestion. A picture of the corresponding normal adjacent colon has been added for comparison (see Figure 5f of revised manuscript).

Reviewer: - Collagen staining for Fig. 6 d+j would be helpful to see if the collagen is actually degraded.

Response: We agree that it is important to verify that collagen is actually degraded by the TEM8 WT TSCs used in the cocultures of Fig. 6. To address this concern, we have now demonstrated that the collagen is internalized and degraded in TEM8 positive TSCs (see Extended Data figures 7d and 8b of revised manuscript).

Reviewer: - Was serum glutamine measured in the TSC/SW620 co-culture to verify the mechanism?

Response: We thank the reviewer for raising this important question. To address this concern, and verify that TSCs are the source of glutamine, we grew TSC-TEM8-WT, TSC-TEM8-KO or TSC-TEM8 WT + m830 antibody on collagen for 48h, to mimic the co-culture assays, and then performed complete amino acid analysis of the cell culture supernatants. As shown in figure 6l of the revised manuscript, the data confirm that glutamine is the only amino acid present in the supernatant of TSC-TEM8 WT cells that is dramatically reduced in the supernatant from both TSC-TEM8 KO cells, and TEM8 WT cells treated with m830 anti-TEM8 collagen blocking antibody, providing strong additional support the proposed mechanism.

Reviewer: - Would a radiolabeling experiment with glutamine make sense to show that the amino acid is actually converted into energy? Or a simple ATP assay (Fig. 6)?

Response: We thank the review for this helpful suggestion and have performed an ATP assay. The data, which are shown in the revised figure 6d and Extended Data figure 12f, demonstrate that addition of glutamine results in more ATP production in the tumor cells.

Reviewer: - Also in Fig. 7: Effect of the antibody therapy on collagen in tumor tissue would be interesting.

Response: While we agree this would be interesting, visualizing an overall buildup of collagen is difficult over the short time frame required for our tumor studies. For example, in select normal organs of TEM8 KO mice, such as the ovaries and uterus, we could only begin to detect an increase in collagen by 4 months, and the increase was much more prominent at 6 months (see PMID: 19622764). One potential reason the buildup is so gradual, is that compensatory feedback mechanisms may slow the production of more collagen in tumor stromal cells when collagen cannot be internalized. Another possibility is that excess collagen can also be degraded by other cell types, such as tumor infiltrating macrophages, which can break down collagen through phagocytosis and other TEM8 independent mechanisms. Regardless, of the mechanisms(s), because the overall buildup of collagen is so slow in TEM8 KO mice, we would not

expect to observe any differences following the relatively short-term antibody treatment.

Reviewer #2 (Remarks to the Author):

In this study, Hsu et al. focus their research on TEM8, they show, using several mouse tumor models, that this cell surface protein, strongly expressed in tumor associated stroma, is crucial for tumor growth. Based on their previous publication showing high expression of TEM8 in stromal cells such as cancer-associated-fibroblasts (CAFs), tumor associated endothelium and pericytes, they describe the essential role of TEM8 in tumor stromal cell (TSCs) for collagen1 uptake and demonstrate that TEM8 expressing TSCs support tumor growth and this pro-tumoral role of TSCs depends on their ability to uptake collagen. In a second part of the manuscript, authors highlight that nutrient stress such as FBS depletion, glutamine restriction, induce TEM8 expression in TSCs, HMECs cells. In these nutrient restricted conditions, addition of collagen1 partly rescues TSCs survival and this rescue is TEM8 dependent. Moreover, authors show that co-culture of TSCs expressing TEM8 with tumor cells is required to observe tumor cell survival upon low FBS expression and collagen addition. From this observation authors suggest that TEM8-dependent collagen uptake by TSCs can further metabolically support cancer cells, through collagen-derived glutamine supply. Finally, using TEM8 specific antibodies to block collagen uptake, authors demonstrate in vivo that TEM8 activity is essential for tumor growth and metastatic burden and that combination of such TEM8 targeting with routine chemotherapy improves the efficacy of several chemotherapeutic agents in several tumor models. Globally this work presents an interesting role of TEM8 in the crosstalk between TSCs and cancer cells and moreover demonstrates that the phenotype observed is common to several cancer types. Also, in this context the determinant role of collagen1 in tumor growth is also highlighted. Finally, authors propose relevant tools to target TEM8 and present strong in vivo validation on mouse tumor models for the use of TEM8 blocking antibodies in clinic. It remains that the demonstration of the metabolic support of TSCs-processed collagen1 to tumor cells is weak and requires solid validation that is not presented in the present manuscript. As authors include a huge amount of data in this manuscript, use several models of tumor mouse models and strongly demonstrate the specific role of TEM8 in tumor growth, I suggest that authors should tone down their conclusions about this metabolic support of col1 and consider the following suggestions/requirements to improve the robustness of their work.

Response: To strengthen our data regarding the metabolic support of TSCs-processed collagen1 to tumor cells, we have now added several additional experiments. We have performed comprehensive amino acid analysis of the supernatants of TSCs which revealed that glutamine is the only amino acid that is lost in TSCs upon genetic disruption of TEM8 or pharmacologic blockade (see figure 6I of revised manuscript). We have also strengthened our knockdown studies by assessing the importance of PRODH in the pathway, and assessed the fate of the glutamine in tumor cells by performing ATP studies. We feel these studies significantly strengthen our conclusion that TSCs-processed collagen provides metabolic support to tumor cells.

Major Comments :

Reviewer: Point 1: Figure1, 2: Some data in these figures show a delay in tumor growth (fig1 a,d,g,h, fig2c,d,f,g,h,j) more than a long term tumor repression, and others show a clear impact on tumor inhibition (fig.1b,e, fig2 k). Hence, authors should give a more robust conclusion in regards with data of these 2 figures:

1/ highlight conclusion about the effect of targeting TEM8 either in fsp+ or VE cad+ cells, as it appears that targeting TEM8 in fsp+ cells is more efficient in term of inhibition of tumor growth than in VE cad+cells,

Response: In every solid tumor model we have analyzed, TEM8 disruption could only slow tumor growth. We have never found TEM8 disruption to completely halt or regress tumors. Using the same tumor model grown in global TEM8 WT or KO mice we have observed some variation in the extent of tumor delay from experiment to experiment. After investigating the basis of this variation, we now think that a major factor that impacts the extent of growth delay is the age of the mice (see revised figure 1g and 1h). However, while TEM8 plays a more prominent role as mice age, it would be impractical to repeat each tumor experiment in mice that are all age-matched, preferably 9 months old. Another technical factor that confounds the comparison is the nature of the cre deleter strains used in our studies. The VE-cadherin cre-ER deleter strain requires tamoxifen for induction, while the Fsp-cre is driven by a constitutive promoter. Tamoxifen induction in the VE-cadherin cre strain is required in order to bypass unwanted deletion in hematologic lineages. Because tamoxifen induced cre-mediated deletion becomes less efficient as mice age, we were compelled to use younger mice for the VE-cadherin studies compared to the fsp-cre studies. Therefore, the impact of TEM8 in VE-cadherin positive cells may be an underestimate. Nevertheless, despite these technical limitations, by evaluating Fsp and VE-cadherin stroma contributions using desmoplastic and non-desmoplastic tumors we can conclude that both TEM8+ vascular cells and TEM8+ CAFs play a critical role in tumor growth promotion. Furthermore, we can conclude that CAFs play a much more prominent role in highly desmoplastic tumors where the CAF content is highest. To address this concern, we have summarized the conclusions drawn from these studies on page 9 of the revised text.

Reviewer: 2/ precisely mention when data show tumor delay vs a decrease in tumor growth,

Response: As mentioned above, TEM8 disruption results in tumor growth delays in all models tested, and the underlying basis of the differences in extent of growth delay is likely to be multifactorial. Comparisons across the tumor models is complicated by the fact that multiple factors can impact the extent of tumor growth delay between tumor studies, including age of the mice, the efficiency of the particular deleter strain, as well as the particular tumor model itself (some tumors may rely more on TEM8 in stroma than others). Therefore, while we remain open to including further discussion if deemed critical, we are concerned that it could draw unnecessary attention away from our main conclusions.

Reviewer: 3/ add some data showing the proliferative index of these tumors (ie their relative aggressiveness by Ki67 staining, especially for panel showing tumor growth delay), 3/ their ECM and col1 content by col1 IHC, trichrome staining and finally H/E staining.

Response: To address this question, we have now added Ki67 staining in PyMT tumors from TEM8 WT and KO mice (see revised Extended Data, figure 4a). We also performed col1 staining of PyMT tumors (see revised Extended Data, figure 4b). We chose to focus on the tumors that develop spontaneously in the PyMT model as these tumors develop over a long period of time and the collagen buildup can be difficult to visualize in fast growing cell line derived tumors (see point 8 from Reviewer 1 for further discussion regarding the gradual buildup of collagen in TEM8 KO mice). The studies in PyMT revealed an overall decrease in collagen 1 in the TEM8 KO tumors. We chose not to perform the trichrome and H&E as these methods stain all collagens and are therefore less-specific. With the PyMT studies, our confocal analysis of the entire tumor revealed a marked variation in Ki67 staining depending on the tumor size

and region evaluated. Therefore, while we could observe a trend towards decreased proliferation in the TEM8 KO, the difference was not statistically significant. Further studies would be required to determine if the trend we have observed represents a real decrease, or if it is just an artifact of the large intratumoral variation. Nevertheless, we feel that such further analysis is beyond the scope of the current study as the main focus of our paper was to determine how the tumor stromal cells exploit collagen degradation pathways to support tumor cell metabolism and ATP production.

Reviewer: In fig2: IHC or IF of FSP+cells in tumors TEM8fl/+ or fl/- would indicate CAFs/TSCs content of these tumors following TEM8 modulation.

Response: While it might be possible to monitor CAF/TSC content following the disruption of TEM8 in FSP+ cells, a very complex cross is required to obtain the TEM8fl/-; Fsp-cre+ or TEM8fl/+; Fsp-cre+ , which would take several months to create. Furthermore, although it is possible that TEM8 deletion in FAP+ cells could theoretically increase or decrease CAF cell numbers in tumors vivo, in either case we would not expect the outcome of this experiment to impact the main conclusions of the paper. Therefore, we suggest that these particular studies are outside the scope of the current study and better suited for future follow-up studies.

Reviewer: Point 2: figure 3: to validate the uptake of Col1 by CHO-TEM8 cells, authors should show quantification of Col1 internalization by FACS, using soluble col1FITC.

Response: We agree that it is important to quantify Col1-FITC uptake into CHO-TEM8 cells using flow cytometry. The requested data can be found in figure 4e and Extended Data 7b of the revised manuscript.

Reviewer: Also, IF showing Col1 internalization in lysosomes using co-if with LAMP1 marker is required. Finally co-IF of TEM8 and Col1 is missing.

Response: Because LAMP1 is not a specific marker of lysosomes in certain cell types (PMID: 31907597), to address this concern we have cultured TSCs with FITC-col and stained the cells with pH sensitive lysotracker Red, a dye which is activated in the acidic environment of lysosomes. The new data, which are shown in Extended Data Fig. 9b of the revised manuscript, reveals clear localization of FITC-col in the lysosomes. Co-localization of TEM8 and col1 is now shown in Extended Data Fig. 9c.

Reviewer: Point 3: Figure 4c-e: As the work is focused on the role of TSCs in collagen uptake and its dependency on TEM8, some of the data showing TEM8-dependent col1 binding, uptake using various TEM8 mutants in panels c-e, should be validated on TSCs.

Response: We agree that TSCs are an important model for our validation work. In order to address the dependency of col1 on TEM8 in TSCs we compared uptake and degradation in TEM8 WT and KO TSCs, and have also assessed uptake following treatment with TEM8 blocking antibodies, or non-specific control antibodies (see Extended Data figures 7d and 8b of revised manuscript). The data demonstrate that TEM8 is the major receptor for collagen I in TSCs. However, these data also reveal that TSC-KO cells can also take up a small amount of collagen in the absence of TEM8, suggesting that TEM8 independent mechanisms also contribute to some low-level collagen uptake in TSCs. For this reason, and because TSCs are much more difficult to transfect than CHO cells, only the CHO system was used for evaluating the impact of the various mutations on collagen uptake.

Reviewer: Figure 4f: Again H/E staining of these tumors is required. Also, IF of wt vs mutant TEM8 would be indicative of its localization in either CAFs or VE cells (co-staining with specific markers of these cells)

Response: As mentioned above (see response to reviewer #1 point 8), we have not found any change in collagen expression in the short-term tumor assays, as the collagen buildup is a gradual process that requires several months to observe. Therefore, we would not expect H/E staining to show us any difference between the MC38 tumors grown in TEM8 WT or TEM8-^{E150V/E150V} KI mice, where the TEM8 in the stromal cells only differs by a single amino acid (E150V). The TEM8 IF staining of MC38 tumors is shown in Extended Data figure 6b of the revised manuscript. The TEM8 E150V mutant protein is expressed normally on the cell surface, which verifies that the protein is folded and processed properly (Extended Fig 10a). Based on this, the TEM8 point mutation is not expected to have an impact on which cell types can produce the protein.

Reviewer: Point 4: Figure 5: Origin of TSCs as to be better defined than just citation of previous work in JCI. Please indicate/show whether these TSCs are TEM8 low, high, mixed as in Fig 5c TEM8 is low at basal expression. Are the TSCsWT used in fig6b different from the ones used in fig 5c and d as levels of TEM8 is different at basal level between these panels? Co-IF of TEM8 and CAFs is required in panel fig5f.

Response: To clarify the origin of the cells, in the text we have now described in more detail how the cells were originally derived (see page 10 of revised text). The level of TEM8 in the TSC-WT cells depends on whether or not the cells have been exposed to stress, such as FBS or glutamine deprivation as shown in Fig. 5. The level of TEM8 detected in TSC-WT cells also depends on how long the Western blot is exposed. In the blot of figure 5, the levels of TEM8 look low at basal conditions because the exposure of the blot was short. This was necessary in order to show the relative increase in TEM8 over time under nutrient starvation. In contrast, in figure 6b (Extended Fig. 7c of revised manuscript) the Western blot was exposed for a much longer time in order to clearly demonstrate that there is no TEM8 protein detectable in the TSC-KO cells, even upon overexposure. To clarify this, we have now modified the methods section (page 33) to make it clear which Western blots were exposed for short (revised Extended Fig 7c) or long (revised Fig. 5) periods. A Western blot comparison of TEM8 levels in TSCs versus tumor cells has also been added to the revised Extended Data figure 12b.

In our previous studies we performed extensive co-localization studies between TEM8 and other stromal cell markers (including CAFs) in human colon cancer (see for example PMID: 29863500). Furthermore, using conditional KO mice we show in figure 2 that multiple TEM8 positive stromal cell types contribute to the tumor activity. Therefore, the goal of figure 5f (now figure 5g of revised manuscript) was not to re-assess which cell types express TEM8, but rather to demonstrate how TEM8 levels in stroma are generally highest near the tumor cell nests. Therefore, we do not expect the percentage of TEM8 positive CAFs in this image to impact the conclusions of our paper. Nevertheless, to address this concern the previously conducted co-localization studies have been highlighted in the text (page 5 revised manuscript).

Reviewer: Point 5: Fig6: In the context of assessing the role of col1 in tumor cell survival upon nutrient stress, authors could evaluate the rescue of col1 addition in condition of low glutamine using dialyzed 10% FBS, to avoid striking reduction of important amounts of nutrient using low FBS.

Response: Because nutrient stress is a key driver of TEM8 expression, nutrient stress is required for the TEM8-dependent responses we are describing. Although TEM8 levels in TSCs also increase somewhat in response to glutamine deprivation, they increase much more dramatically in response to low FBS (compare figures 5c and 5e). Glutamine is a non-essential amino acid that only becomes conditionally essential under catabolic stress conditions, for example, when FBS levels in the media are reduced. Therefore, addition of 10% serum allows cells to manufacture their own glutamine for energy production, alleviating much of the stress imposed by low glutamine. In tumors, cells that are far removed from the vascular supply experience stress due to the simultaneous depletion of both small molecular weight (glutamine) and large molecular weight (growth factors) molecules. However, because diffusion in tumors decreases with increasing molecular weight, growth factors, due to their larger size, are even more limited with increased distance from blood vessels. For these reasons we think that FBS depletion is more relevant for the phenotypes being explored.

Reviewer: Blots showing levels of TEM8 in SW620 should be shown in this manuscript. This would add precision to the sentence “although cancer cell TEM8 status did not impact these responses”.

Response: We agree and have added the requested data to the revised Extended Data Fig. 12b.

Reviewer: In panel fig6e data showing SW620 viability when co-cultured with TSCs in low FBS without addition of col1 would indicate the rescue effect of TSCs only on cancer cells in condition of low FBS and this point would allow comparison with the same condition with addition of Col1, in order to evaluate 1/the rescue of TSCs alone on cancer cell survival at low FBS and 2/the effect of col1 on cancer cell survival in condition of co-culture.

Response: If we understand the question correctly, the reviewer is asking what impact TSC-WT and TSC-KO cells have on SW620 tumor cell viability in the co-cultures in the presence versus absence of added exogenous collagen. To address this point, we have added a new experiment (Extended Data Fig. 13a) to the revised manuscript. The data show that WT-TSCs rescue SW620 cells from collagen 1 toxicity better than KO-TSCs.

Reviewer: Point 6: As prolidase specifically allows proline release from dipeptides of proline-glycine from collagen, and GLUL produces glutamine from glutamate, neither PEPD nor GLUL are enzymes in the collagen degradation pathway nor they are key enzymes to be targeted to assess if col1 is converted into glutamine (please re-phrase the associated sentence in the manuscript).

Response: We agree and thank the reviewer for this comment. To clarify we have modified the text to make it clear that these are key enzymes known to be involved in the conversion of collagen degradation products into glutamine (page 16 of revised manuscript).

Reviewer: Hence, fig 6h and I data only show that proline release and glutamine production in TSCs cells support SW620 survival on collagen plates at low FBS and that only glutamine rescues the KO of both enzymes. Based on this observation, and on data from fig 6j, as glutamine from TSCs might derived from various metabolites other than proline, authors definitively have to KO PRODH enzyme to inhibit glutamine production from proline degradation to conclude that gln release from TSCs collagen-derived proline supplies cancer cells with glutamine pool (use of PRODH KO TSCs co-cultures +/-col1, +/-gln/pro). Another way to demonstrate this would be proline tracing in TSCs cultures to follow carbon flux from proline derived col1 to glutamine and supply this labelled gln to cancer cells to assess the fate of these proline-derived carbon into cancer cells.

Response: To address this concern we have taken two approaches. First, we have used lentiviral particles to knock down PRODH in TSCs by employing two different shRNAs for the co-culture experiment, using the same shRNA knockdown strategy that was used for PEPD and GLUL in the manuscript. Second, we also created a gRNA lentiviral vector and introduced it along with CRISPR/Cas9 into TSCs to create PRODH knock-out TSCs. As shown below, both the knockdown (KD) and knockout (KO) of PRODH using viral particles reduced PRODH levels similarly (Western blot, left panel) and resulted in low SW620 cell viability, an effect that could be reversed by adding exogenous glutamine (co-culture assays, right panels). These results provide strong additional evidence for the importance of glutamine derived from collagen in promoting tumor growth. Because the KD and KO produced comparable results, in the manuscript we have also added the KD data, which employed the same method that was used for PEPD and GLUL.

Reviewer: Point 7: figure 7: the effect of TEM8 on metastasis is surprising as metastasis microenvironment contains few and variable amounts of CAFs and have abundant supply of oxygen and nutrients as compared to primary tumors. Hence this raises questions about the role of TEM 8 in metastases as a driver of collagen-derived metabolic support. Data showing TEM 8 staining in metastasis are missing.

Response: While we agree that there can be some variability in the amount of CAFs present in colon cancer liver metastases, we have found that there are a lot of CAFs in these tumor metastasis. Indeed, this was a major reason we elected to test the role of TEM8 neutralizing antibodies in colon cancer liver metastasis. Others have also estimated that cancer-associated fibroblasts (CAFs), which constitute the main stromal cell type in colon cancer liver metastases, can account for up to 75–80% of the entire tumor mass (PMID: 33546502). To address this issue, we have now added TEM8 and collagen I staining of human colon cancer liver metastasis, as well as the HCT116 and MC38 colon cancer liver metastasis models used in our paper (Extended Data Fig. 15 of revised manuscript). The data reveal the high levels of both TEM8 and collagen I in these tumors, which may help explain why these tumor metastasis are sensitive to TEM8 antibody therapy. Others have also shown that the supply of oxygen (HIF-1-alpha staining) is similar between primary colon tumors and colon cancer liver metastases and that higher levels of HIF-1 α expression in liver metastasis (i.e., lower oxygen) predicts a worse prognosis (PMID: 32269630 and 32895059).

Minor points :

Reviewer: Minor Point 1: Fig1a-c: precisions of athymic nude mice models have to be added in mat and

met section. Numbers of mice in groups of these panels have to be indicated as for other panels where it is missing.

Response: In the revised text under the “Animals” heading of the material and methods section, we have highlighted the original paper which describes the generation of the TEM8 KO mice on an athymic nude background (PMID: 22340594). The number of mice per group in Fig. 1a-c have now been added to the revised manuscript.

Reviewer: Minor Point 2: Panel fig 1g should be placed in sup data. Panel fig 2i shows only UACC tumor not SW620, RENCA as mentioned in the text: is there a panel missing in this figure or in sup data?

Response: Fig 1g has now been placed in supplemental data as requested (Extended Data fig. 5). We thank the reviewer for noticing the discrepancy between the text and the figure. The text has been corrected.

Reviewer: Minor Point 3: Stats: indicate in figure legends to which comparison p value are associated with.

Response: We have modified the figure legend to clarify that the p-values represent a comparison between the TEM8 WT and KO group at each given time point post tumor inoculation.

Reviewer #3 (Remarks to the Author):

Kuo-Sheng Hsu et al used mouse models and in vitro experiment to investigate the role of TEM8 in tumor development and cancer cell survival. The findings are interesting, results are convincing and controls experiments well performed. The manuscript lacks overall methods description and reagents references.

Reviewer: 1. The authors use a TEM8 KO model in this manuscript: do the TEM8 KO mice show any phenotype like head or dental malformation as in GAPO patients? Same for conditional KO mice (Tie2-cre, VE-cad-CreERT2 and Fsp1-Cre)?

Response: We assume the reviewer is asking about the phenotypic abnormalities of the new TEM8-E150V knockin (KI) mice, as the phenotype of the global TEM8 KO mice was previously described and shown to be similar to that in GAPO patients (see PMIDs: 19622764 and 23602711). It has been proposed that each of the phenotypes caused by TEM8 loss is a result of excess collagen buildup (see PMIDs: 19622764, 23602711 and 2248288). Importantly, the TEM8 KI mice also share the same phenotypic abnormalities as the KO mice (growth retardation, dental malformations and frontal bossing) and the new data has been added to the revised manuscript (Extended data Fig. 11). Although it would be interesting to determine which TEM8 positive cell types are important for these phenotypes using conditional KO mice, we feel this is outside the scope of the current study which focuses on the role of TEM8 in tumor growth promotion. Therefore, we feel those studies are better suited for an independent follow-up study.

Reviewer: 2. General use of collagen in the description of the results should be replaced by the appropriate collagen depending of the experiment performed. For example, page 7 (Figure 1f) the authors claim “increased collagen levels in tumors derived from TEM8 KO versus TEM8 WT animals” Please specify the collagen type (Collagen I only shown in this figure).

Response: For figure 1f, a collagen binding probe called CNA35 (PMIDs: 25490719) was used for the staining. This collagen binding probe has been shown to bind collagens type I, III and IV. (PMID: 16476406). We have now clarified this in the text and methods sections, and indicated the known collagen in places where it is known.

Reviewer: 3. In figure 1h,i, the authors compare the percentage of tumor growth inhibition between WT and KO animals at 4 and 11 months. However, the comparison does not seem to be done at the same moment (Different day after inoculation) Please clarify or compare similar dpi please.

Response: We thank the reviewer for raising this concern. Because MC38 tumors grow faster in the younger mice, some of the tumors reached the maximum allowable size earlier in the younger cohort, such that those studies ended earlier (day 18 instead of day 21). However, we agree that it is best to compare tumor growth inhibition (TGI) at the exact same time point. Therefore, to address this concern we have now added the TGI for the 18-day time point for the older mice (Revised figure 1g,h). Because the %TGI at day 18 (76%) is similar to what it was at day 21 (78%), this does not impact our conclusion that the difference in tumor growth is higher in old compared to young mice.

Reviewer: 4. Figure 2: On page 9 the authors claims than TEM8+ vascular cells in the SW620 and RENCA models play minor roles but no data are shown for this. Please clarify.

Response: The evidence that vascular cells play no role or only a minor role in the tumor phenotype is based on the fact that TEM8 disruption using our fibroblast (fsp) deleter, which does not delete in vascular cells, results in a tumor growth inhibition that is similar to that observed in global TEM8 KO versus WT mice (compare RENCA and SW620 in Fig. 2e with RENCA in Figs. 1d and SW620 in Fig. 2G of PMID: 22340594). Furthermore, CAF levels are very high in RENCA and SW620 compared to vascular cells (see Extended Data Fig. 6b). To address this concern, we have modified the text to clarify the evidence implicating the role of fibroblasts in these models. Although challenging to measure, the small fraction of TEM8 positive vascular cells in these models could theoretically contribute in a minor way to the tumor phenotype observed. We have modified the text to clarify that these cells may play either no role, or only a minor role compared to TEM8 positive CAFs.

Reviewer: 5. The authors describe the presence of laminin and collagen IV in matrigel, can they provide a reference, do they know the amount of Collagen 1 or Collagen VI in Matrigel.

Response: A reference describing the composition of Matrigel has been added to the revised text. According to the manufacturer, Matrigel is composed of approximately 60% laminin, 30% collagen IV, and 8% entactin, while collagen 1 is present only in trace amounts (see <https://www.corning.com/catalog/cls/documents/faqs/CLS-DL-CC-026.pdf>, and PMID: 20162561).

Reviewer: 6. figure 3 bc, the amount and origin of collagens are described only for Collagen I, are the other collagens used at 200ug/ml? could the authors give references for collagen they used.

Response: The concentration of collagen varied depending on the assay and whether or not the collagen was coated onto plates or polymerized into a gel. The ECM concentrations have been added to the revised manuscript.

Reviewer: 7. figure 3h, could authors give the concentration of the gradient of COL1 and COL6 they are using. Molecular weights markers should be added.

Response: The concentration gradient has been added to the figure legend, and the MW markers to the figure.

Reviewer: 8. The authors use TEM8+ and TEM8- (and sometimes CMG2+ cells) to describe CHO overexpressing or not TEM8 or CMG2. Please use the correct name CHO, CHO-TEM8 (for CHO overexpressing TEM8) or CHO-CMG2 (for CHO-overexpressing CMG2). Showing endogenous levels of CMG2 and TEM8 in CHO could be useful.

Response: We have updated the text using the suggested terminology which we agree adds clarity. In the text and methods, we have now also referenced our previous paper which shows the relative levels of CMG2 and TEM8 in the parental CHO (CHO-PR230) cells, which lack TEM8 and CMG2, as well as the TEM8 or CMG2 overexpressing sublines.

Reviewer: 9. Figure 3e : the binding assay is not described in the material and methods. How is it performed?

Response: We thank the reviewer for noticing our accidental exclusion of this assay. The cell binding assay protocol has now been added to the material and methods section.

Reviewer: 10. figure 4-a : D50, S54, T118, Q137 should also be included in the alignment among species, with E152 and H154.

Response: As requested, we have included in the revised figure 4a an alignment the D50, S54 and T118 mutations that are highly conserved and involved in coordinating the metal ion at the center of the MIDAS motif. In the case of ITGA2, the metal ion has been shown to directly bind collagen. The location of the Q137L mutation, which was derived from a patient with GAPO syndrome, was not included because in the TEM8 crystal structure this amino acid is located at the bottom of the I-domain, far removed from the collagen binding MIDAS domain. Therefore, its ability to block collagen binding is most likely indirect, through allosteric regulation of the MIDAS. While we examined Q137V in our collagen binding assays to verify the importance of TEM8-collagen binding in GAPO, it was not considered further in our study due to its complex mechanism of action.

Reviewer: 11. Figure 4d, degradation of collagen is quantified in the culture supernatant but material and methods only describe IF staining. Please explain methods.

Response: The requested methods can be found under the heading "Soluble FITC Assay" in the revised methods.

Reviewer: 12. figure 5abcde: Molecular weights markers should be added.

Response: Molecular weight markers have been added to the Western blots of figure 5 as requested.

Reviewer: 13. Supplementary Figure 2: no description of arrow in the legend.

Response: The description of the arrowheads in supplementary Fig. 2 (Extended Data Fig. 6a of revised manuscript) have been added to the figure legend.

Reviewer: 14. In some figure legends, the number of animals/replicates used is missing.

Response: We thank the reviewer for noticing this omission. The missing numbers have been added to the figure legends.

Reviewer #4 (Remarks to the Author):

In this study by Usu et al, the authors describe a novel mechanism regulating tumor growth through the uptake of glutamine released by fibroblasts upon collagen uptake and processing into glutamine. This is a very interesting study with substantial amount of data supporting the main hypothesis and multiple mouse models to verify that TEM8 in the tumor stroma regulates tumor growth through collagen consumption. The data presented open a new avenue of investigation into how regulation of the ECM by the stroma control tumor growth.

There are some aspects that could improve the study. Below I highlight my comments:

Reviewer: 1) In figure 1, the authors show that in aged mice the effect of TEM8 KO in the stroma is larger than in younger mice. Have the authors tested if TEM8 decreases also with age?

Response: We thank the reviewer for this interesting question. Our study to compare tumor growth in mice of different ages was part of a larger effort to identify what factors influence the extent of tumor growth delay observed in TEM8 WT versus KO mice, which we had noticed varied somewhat from experiment to experiment. While we did not observe any obvious impact of aging on TEM8 staining histologically, more quantitative assays would be required to determine if TEM8 levels were altered subtly. While such studies could be informative, we do not think they would impact the main conclusions of our paper, and such a rigorous analysis would require us to start these 9-month studies over again. For these reasons we think the suggested studies are better suited for a follow-on study focused on understanding TEM8 function in old versus young mice.

Reviewer: 2) Figure 2 a,b is a little confusing. It's my understanding that these images are control of the Tie2-Cre, VE-cad-cre and Fdp1-cre mice but the way it's written in the text gives the understanding that these images are from mice with TEM8 deleted in those compartments. The authors should show images of TEM8 staining in the mouse models upon TEM8 KO in the different compartments to verify depletion.

Response: We thank the reviewer for noticing this and apologize for the original text which indeed was confusing. The reviewer is correct, these mice were only used as a control to ensure the fidelity of the cre deleters prior to using them for our conditional TEM8 KO studies. We have now modified the text to make it clear that the reporter mice were used only as a tool to evaluate where cre was being expressed. We would not necessarily expect deletion of TEM8 to impact stromal driven cre expression, and the cross required to combine the two transgenes (cre and mTmG) with two TEM8 null alleles would take 6 to 8 months to perform. Therefore, this cross was never actually performed even though our original text (prior to correction) made it sound like it may have been performed.

Reviewer: Also, the images shown do not have scale bar.

Response: The missing scale bar has been added.

Reviewer: In this figure the tumor growth panels are organized in a way that makes the figure difficult to understand. I will suggest to group the graphs in blocks, i.e. all the FSp-cre (e,h,j,k) in the different cell lines under the same letter; for example (c). I think this will help understand the figure.

In figure 2i the authors mention in the text that SW620, RENCA and UACC models are shown but only UACC graph is shown in the figure.

Response: The images have been reorganized as recommended which we agree helps makes the figure easier to follow. We have also corrected the text so that it matches what is shown in the figure.

Reviewer: 3) In figure 3 the authors show that TEM8 binds to COL I and COL VI. Does TEM8 bind to other fibrillar collagens (II,III) or it's restricted to collagen I?

Response: In our manuscript we tested three different collagens. However, because there are at least 28 different collagen types including 7 fibrillar collagens (types I, II, III, V, XI, XXIV and XXVII) (PMID: 28101870), we feel that a comprehensive analysis of all collagen types that bind TEM8 is beyond the scope of the current manuscript. While TEM8 may indeed bind other collagens, we do not expect this will alter the conclusions of our paper as collagen I is by far the most abundant collagen, and collagens account for 25% of total proteins in the human body.

Reviewer: 4) The authors use extensible CHO cells as a model of low TEM8 but later in the manuscript the authors show data with fibroblasts either WT or KO for TEM8. This seems to be a more representative model for the study the authors propose. Using these fibroblasts lines in the experiments described in Figure 3 and 4 will strength the paper.

Response: We agree that the TSCs are a more physiologically relevant model for these studies. However, the CHO parent cells are unique in that they are relatively non-adherent and lack both TEM8 and collagen binding integrins. Another problem with the TSCs is that these cells are extremely difficult to transfect with high efficiency, complicating an analysis of the TEM8 mutants in this system. To address this concern, we have now performed the FITC-collagen uptake and degradation assays in the TSC-WT and TSC-KO cells (Extended Data figs. 7d and 8b of revised manuscript). We have also shown that our TEM8 neutralizing m830 antibody can block the FITC-collagen uptake into TSC-WT cells (Extended Data fig. 7d). While the data implicate TEM8 as the major collagen I receptor in TSCs, a small amount of collagen I uptake in TSC-KO cells indicates that there is also a small amount of TEM8-independent uptake in these TSCs.

Reviewer: It will also good to show a western blot of TEM8 in CHO and CHO-TEM8 cells to assess the level of overexpression.

Response: The levels of TEM8 in CHO and CHO-TEM8 have been added to the revised manuscript (Extended Data Fig. 7a).

Reviewer: 5) In figure 3 and supp3 several experiments showing degradation of FITC-Col I are presented however no quantification is shown.

Response: We agree that quantification of FITC-Col I degradation is important. To quantitatively evaluate the degradation of FITC-coll we employed two established assays – one involved growing the cells on top of a gel of FITC-Col for 48h and then using ELISA to measure the FITC that is realized into the supernatant following FITC-coll uptake and degradation. This assay takes advantage of the fact that FITC is hydrophobic, such that upon release from collagen inside cells it diffuses freely through cell membranes into the extracellular media. The second assay involves flow cytometry to quantitatively measure the FITC-coll trapped inside cells after internalization but before it is degraded. The data from these assays can be found in Fig. 4d,e, Fig. 7b, and Extended Data Fig. 7b,d and 10b of the revised manuscript. We have also modified the text to highlight how these quantitative assays were performed.

Reviewer: 6) The metastasis experiments are interesting. However, performing those experiment in a spontaneous metastasis model will strength the claims. In the paper the authors use intrasplenic injection to deliver colon cancer cells to the liver. But, how is the metastatic cascade affected in the presence of m830? Does TEM8 blockage prevent dissemination and seeding of tumor cells in the liver? Does it prevent tumor growth at the metastatic sites? Experiments evaluating the metastasis potential of m830 in spontaneous metastasis model will strength this part of the paper.

Response: Our current studies point towards a major role for TEM8 in promoting the growth of pre-established metastases as they grow and deplete the nutrients locally, rather than promoting the establishment of metastases *per se*. We chose to evaluate established colon cancer liver metastases because we found that these established metastatic lesions in humans express a high level of TEM8 and collagen (data now shown in Extended Data Fig. 15a of revised manuscript). In order to generate colon cancer liver metastasis, tumor cells were seeded via intrasplenic injection, as this is the most widely used and robust assay for creating widespread established liver metastases. Based on the mechanism we have uncovered we do not necessarily expect TEM8 function blocking antibodies will prevent dissemination and seeding of metastasis. While this is an interesting question, our studies point towards a role for TEM8 in promoting tumor growth by providing tumor cells with a source of exploitable fuel (glutamine) under metabolic stress conditions. The metabolic stress we are describing develops in solid tumors as tumor cells rapidly divide and outstrip their blood supply (which is why tumor angiogenesis is critical for expansive tumor growth). Because recently disseminated tumors cells are highly exposed to growth factors, nutrients and glutamine from the circulation, we would not expect such recently lodged tumor cells (micrometastatic disease) to require an alternative source of energy such as collagen-derived glutamine. In other words, it is only after the tumors have become established that we would we expect them to deplete nutrients locally. Furthermore, we are not aware of spontaneous colon cancer liver metastases models that are highly tractable and would allow us to monitor long-term antibody treatment in immunodeficient mice. Therefore, we acknowledge the possibility that TEM8 may have additional uncharacterized independent function(s) in promoting the dissemination and seeding of metastasis. However, due to the challenging nature of such work and the fact that we do not expect the outcome to impact the conclusions of the current work, we consider this a separate question outside the scope of the current study. Nevertheless, we agree that this line of investigation would be worthwhile pursuing in future independent follow up studies.

REVIEWERS' COMMENTS

Reviewer #1 (Remarks to the Author):

The authors have addressed the concerns in depth and with additional experiments.

Just one aspect that became apparent in the answers of the authors: e.g. immune cells can also digest collagen, and it could be that after the initial success of the antibody therapy, the immune cells ultimately the effect because they take over the task of the TSC (see FIG. 6a).

Reviewer #2 (Remarks to the Author):

Authors have answered as best as they could my concerns and major points.

However, as there is no metabolic tracing of collagen to clearly demonstrate that glutamine derives from collagen, I would recommend to avoid the terms "collagen-derived", in the sentence : « tumor cells exploit collagen-derived glutamine » in the introduction. Please rephrase this sentence.

Minor details : typo correction : correct micropinocytosis->macropinocytosis

When not possible or not appropriate to perform the new experiments I required, authors clearly explain their reasons and their scientific arguments. Therefore I consider the manuscript ready for publications, and thank the authors for their efforts to improve their manuscript.

Reviewer #3 (Remarks to the Author):

The authors have satisfactorily addressed our concerns. This is a very nice piece of work!

Reviewer #4 (Remarks to the Author):

I would like to thank the authors for answering my questions from the previous version.

The authors did a great job addressing my comments and concerns.

The authors strength the study in this new revised manuscript.

I don't have any other further comments.

Congratulations on this nice work.

Point-by-point response to the reviewers' comments

REVIEWER COMMENTS

Reviewer #1 (Remarks to the Author):

Reviewer: The authors have addressed the concerns in depth and with additional experiments. Just one aspect that became apparent in the answers of the authors: e.g. immune cells can also digest collagen, and it could be that after the initial success of the antibody therapy, the immune cells ultimately the effect because they take over the task of the TSC (see FIG. 6a).

Response: We thank the reviewer for their positive response to our revision and all their constructive comments. Although we do not have any evidence for immune cells as yet, it is theoretically possible that immune cells contribute to the TEM8 antibody responses we have observed. However, this is difficult to address as we have found that the immune cells in tumors, especially macrophages, are made up of many subpopulations, which complicate the analysis of their role. As such, we feel these studies are better suited for long-term follow-up studies.

Reviewer #2 (Remarks to the Author):

Reviewer: Authors have answered as best as they could my concerns and major points. However, as there is no metabolic tracing of collagen to clearly demonstrate that glutamine derives from collagen, I would recommend to avoid the terms "collagen-derived", in the sentence : « tumor cells exploit collagen-derived glutamine » in the introduction. Please rephrase this sentence. Minor details : typo correction : correct micropinocytosis->macropinocytosis When not possible or not appropriate to perform the new experiments I required, authors clearly explain their reasons and their scientific arguments. Therefore I consider the manuscript ready for publications, and thank the authors for their efforts to improve their manuscript.

Response: We thank the reviewer for their positive response to our revision and all their constructive comments which have strengthened the manuscript. We have modified the sentence in the introduction, which we agree helps to more accurately reflect the data provided. We also thank the reviewer for catching the typo which has now been corrected.

Reviewer #3 (Remarks to the Author):

Reviewer: The authors have satisfactorily addressed our concerns. This is a very nice piece of work!

Response: We thank the reviewer for their constructive comments during the review process and their strong endorsement of our work.

Reviewer #4 (Remarks to the Author):

Reviewer: I would like to thank the authors for answering my questions from the previous version. The authors did a great job addressing my comments and concerns. The authors strength the study in this new revised manuscript. I don't have any other further comments.

Congratulations on this nice work.

Response: We thank the reviewer for their constructive comments during the review process and their strong endorsement of our work.